# NPAS4 supports cocaine-conditioned cues in rodents by controlling the cell type-specific activation balance in the nucleus accumbens

Brandon W. Hughes [1,4], Jessica L. Huebschman[1,4], Evgeny Tsvetkov[1], Benjamin M. Siemsen[2], Kirsten K. Snyder[1], Rose Marie Akiki[1,3], Daniel J. Wood[1,3], Rachel D. Penrod[1], Michael D. Scofield [2], Stefano Berto [1], Makoto Taniguchi [1] ✉ & Christopher W. Cowan [1] ✉

Powerful associations that link drugs of abuse with cues in the drug-paired environment often serve as prepotent relapse triggers. Drug-associated contexts and cues activate ensembles of nucleus accumbens (NAc) neurons, including D1-class medium spiny neurons (MSNs) that typically promote, and D2-class MSNs that typically oppose, drug seeking. We found that in mice, cocaine conditioning upregulated transiently the activity-regulated transcription factor, Neuronal PAS Domain Protein 4 (NPAS4), in a small subset of NAc neurons. The NPAS4+ NAc ensemble was required for cocaine conditioned place preference. We also observed that NPAS4 functions within NAc D2-, but not D1-, MSNs to support cocaine-context associations and cue-induced cocaine, but not sucrose, seeking. Together, our data show that the NPAS4+ ensemble of NAc neurons is essential for cocaine-context associations in mice, and that NPAS4 itself functions in NAc D2-MSNs to support cocaine-context associations by suppressing drug-induced counteradaptations that oppose relapse-related behaviour.

A fundamental neurobiological mechanism underlying relapse vulnerability in substance use disorder (SUD) is the formation of powerful associations between drug-induced euphoria and the diffuse and discrete cues in the drug-use environment[1–3]. The nucleus accumbens (NAc) represents a key brain region that regulates motivated behavior, including relapse to drug seeking, and in animal models of SUD, drug experience-dependent neuronal plasticity in this region is required for future drug-seeking behavior[4,5]. The NAc comprises a large population (>90%) of GABAergic long-range projection neurons, commonly referred to as medium spiny neurons (MSNs) or spiny projection neurons[6]. MSNs are further divided into two major subpopulations that express either D1-class dopamine receptors (D1-MSNs) or D2-class dopamine receptors (D2-MSNs), which integrate motivational and reinforcement-based information from cortical and subcortical areas

to facilitate goal-directed behaviors and drug seeking[7]. In rodents, drug seeking is often studied using the drug conditioned place preference (CPP) assay or the intravenous drug self-administration (SA) assay. CPP measures the formation of drug-context associations following experimenter-delivered (i.e., non-contingent) drug, whereas drug SA, followed by extinction training and reinstatement tests, models active drug use and relapse-like behavior. While many brain regions and circuits are involved in these complex addiction-related behaviors, the activation of glutamatergic inputs from the prelimbic prefrontal cortex (PFC) to the NAc core (PFC→NAc) is an essential circuit involved in cue-reinstated cocaine and heroin seeking[8–13].

Activation of NAc D1-MSNs promotes drug reward-context associations and drug-seeking behavior; whereas, activation of D2-MSNs typically opposes these behaviors[13–23]. Indeed, optogenetic activation

[1]Department of Neuroscience, Medical University of South Carolina, Charleston, SC, USA. [2]Department of Anesthesiology, Medical University of South Carolina, Charleston, SC, USA. [3]Medical Scientist Training Program, Medical University of South Carolina, Charleston, SC, USA. [4]These authors contributed equally: Brandon W. Hughes, Jessica L. Huebschman. ✉e-mail: taniguch@musc.edu; cowanc@musc.edu

of D2-MSNs inhibits drug seeking by suppressing the activity of downstream brain regions, such as the dorsolateral ventral pallidum, through feedforward inhibition, while D1-MSN activity promotes cocaine-seeking behaviors[14,24–26]. These activated D2-MSNs can also directly inhibit D1-MSNs by lateral inhibition within the local NAc microcircuit to suppress drug-related behavior[14,27–29]. To add to the complexity of D1- and D2-MSN-mediated drug seeking, activity in a subpopulation of neurons that induces the immediate-early gene, *Fos*, during drug exposure is required during the future expression of many drug-related behaviors[30–32]. Although FOS is often used as a marker of recent neuronal activity, it can also be induced by other stimuli, including neurotrophins and growth factors[30,33,34]. In contrast, the calcium-dependent immediate early gene (IEG) and transcription factor, neuronal PAS domain protein 4 (NPAS4), is selectively activated by synaptic activity and L-type voltage-gated calcium channel signaling in neurons[34–38]. NPAS4 in forebrain neurons maintains microcircuit homeostasis; in pyramidal neurons, it is reported to enhance inhibitory synaptic transmission, while in somatostatin-positive interneurons, it augments excitatory synaptic transmission[34,36–41]. However, the role of NPAS4 in the regulation of striatal circuits and SUD-related plasticity and behavior is poorly understood. NPAS4 is induced rapidly and

transiently in a small subset of NAc neurons during cocaine conditioning, and its NAc expression is required for cocaine CPP[42] and cocaine locomotor sensitization[43], but also see data herein and[42]. Therefore, understanding how NAc NPAS4 regulates cocaine-context/cue associations and cocaine-seeking behaviors provides new insights into NAc adaptations that shape SUD-relevant behaviors.

## Results

### Cocaine conditioning-induced NAc Npas4+ ensembles are required for CPP

To test the role of the small population of NAc NPAS4-inducing neurons during cocaine conditioning on drug-context associations, we generated and validated a Targeted Recombination in Activated Populations (TRAP; similar to ref. 44) knock-in mouse with tamoxifen-sensitive Cre recombinase expressed with the endogenous NPAS4 coding sequence as a cleavable fusion protein under the control of the *Npas4* gene (i.e., NPAS4-P2A-Cre-ERT2 or NPAS4-TRAp2a) (Figs. 1A and S1A, B). Following infusion of a neurotropic virus expressing Cre-dependent inhibitory DREADD (AAV2-DIO-hM4Di-mCherry) or negative control (AAV2-DIO-mCherry) into the NAc of NPAS4-TRAp2a mice (Figs. 1A and S1C), we administered tamoxifen (4OHT, 20 mg/kg; i.p.)

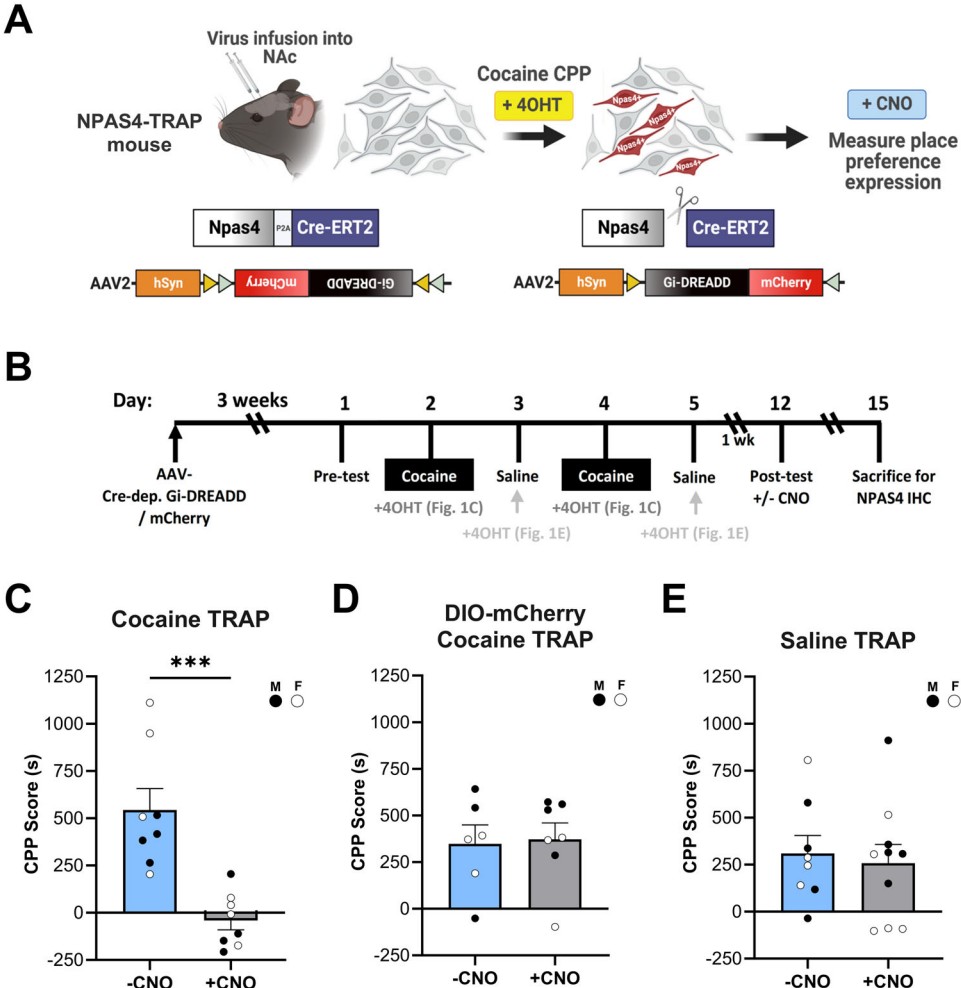

**Fig. 1 | Cocaine conditioning-induced NAc Npas4+ ensemble is required for CPP.**
**A** Experimental design for Gi-DREADD-mediated manipulation of the NAc Npas4+ ensemble using newly generated NPAS4-TRAP mice. **B** Experimental timeline for cocaine CPP, including the timing of 4OHT injections for both cocaine conditioning and saline conditioning TRAP experiments. **C** Inhibition of the cocaine-conditioned NPAS4-TRAP population during the post-test significantly decreases CPP expression ($n = 8$ mice/group; unpaired $t$-test, $t = 4.688$, df = 14, $p = 0.0003$). **D** CNO

administration in animals injected with a control virus lacking hM4Di has no effect on cocaine CPP expression ($n = 6$ and 7 mice/group). **E** Inhibition of the NPAS4 ensemble captured during saline conditioning sessions also has no effect on cocaine CPP expression ($n = 8$ and 10 mice/group). CPP scores are calculated as time in the cocaine paired chamber minus time in the unpaired chamber during the post-test. Pre-test scores are shown in Fig. S1. Data are shown as mean ± SEM; ***$p < 0.001$. Source data are provided in the Source Data file.

immediately following either cocaine (7.5 mg/kg, i.p.) or saline (0.9%, w/v) conditioning sessions to induce stable expression of Gi-DREADD (or mCherry control) in the Npas4+ neurons (Fig. 1B). Injection of clozapine-N-oxide (CNO, 5 mg/kg; i.p.) to inhibit the cocaine-TRAPed ensemble during the post-test significantly reduced cocaine CPP in Gi-DREADD-expressing animals (Figs. 1C and S1D), with no effect observed in mCherry-only controls (Figs. 1D and S1E). When 4OHT was administered following saline, rather than cocaine, conditioning sessions, a similar number of NAc neurons were recruited to the Npas4+ ensemble (Fig. S1G), presumably encoding information about the saline-paired context. However, Gi-DREADD-mediated inhibition of this saline-TRAPed ensemble had no effect on cocaine CPP (Figs. 1E and S1F). No differences were observed in total locomotor activity during the post-test session for either the cocaine- or saline-TRAP experiments (Fig. S1H, I), suggesting that our CPP findings were not due to unanticipated effects of CNO administration on general activity levels. Notably, we also observed that ~40% of the Npas4+ cells TRAPed during the initial two sessions of either cocaine or saline conditioning re-induced NPAS4 expression (as detected by immunohistochemistry) during an additional, third conditioning session (Fig. S1J), suggesting that a subset of the cells activated during the initial context conditioning were reactivated with repeated exposures.

## NPAS4 is induced by cocaine conditioning predominantly in medium spiny neurons

We crossed the NPAS4-TRAp2a mice with the reporter line, Ai14 (R26-LSL-tdTomato) and administered 4OHT during two cocaine conditioning sessions and then performed in situ hybridization with RNAscope probes for dopamine D1 or D2 receptor mRNAs (Drd1a and Drd2, respectively). We observed that 76% of NAc NPAS4-TRAPed neurons expressed either Drd1a (49%) or Drd2 (27%) (Fig. 2A). Importantly, NPAS4-TRAp2a:Ai14 animals that did not receive tamoxifen possessed one or fewer tdTomato-positive cells in the NAc (Fig. S2A), confirming the high fidelity of tamoxifen-dependent recombination in this new TRAP line. Using D1-tdTomato x D2-eGFP reporter mice and immunohistochemistry, we observed approximately 49% overlap of NPAS4 protein-positive neurons with either tdTomato or eGFP after cocaine conditioning (Fig. S2B). Similar to the RNAscope results, we detected more NPAS4+ D1-tdTomato cells (~60%) than NPAS4+ D2-eGFP cells (~40%), and we detected an NPAS4+ population that was negative for both tdTomato (D1) and eGFP (D2).

To fully characterize the NAc cell populations that express Npas4, we performed single-nucleus RNA-sequencing (snRNA-seq) of NAc tissue isolated from wild-type (WT) mice either directly from the home cage or 15 min after a single cocaine conditioning session (7.5 mg/kg, i.p.) (Fig. 2B), a time point previously shown to represent peak Npas4 mRNA expression[42]. Based on quality control stringency and unsupervised clustering (Fig. S2C–E), we defined and analyzed NAc cell-type clusters, including subgroups of D1- and D2-MSNs, interneurons (e.g., Pvalb+, Sst+, ChAT+), and non-neuronal cells such as microglia, astrocytes, and oligodendrocytes (Fig. 2C and Supplementary Data 1). Of note, the observed interneuron population in this experiment was higher than generally reported for the NAc and is largely driven by a relatively high number of Pnoc+ neurons. It's possible that this cluster arises from BNST contamination, where Pnoc+ cells are highly expressed[45]. Regardless, all quantitative comparisons were made within experiment, relative to the sampled population, to help account for the variability that often arises across single-cell sequencing experiments. We also observed a small, non-canonical MSN cluster (MSN_Grm8+) associated with Grm8 (metabotropic glutamate receptor 8) expression, as previously reported by several studies[46–49]. These clusters were defined by specific, previously published gene cluster markers of NAc cell types, such as the enrichment of Drd1 in three subclusters (Fig. 2C, D)[49–51]. Npas4 mRNA expression was detected almost exclusively in neurons, with ~8.5% of the total neuronal

population in either the home cage control or cocaine conditioning groups identified as Npas4+ (Fig. 2E, F). We observed a similar distribution of Npas4+ cell types in each group (i.e., 40% D1-MSNs, 25% D2-MSNs, 21–27% interneurons, and 5–8% D3- or Grm8-MSNs), a distribution similar to the total neuronal population sampled in this experiment. While these data indicate the presence of a similar number of Npas4 mRNA-positive nuclei between the groups, we expected that cocaine conditioning would induce higher Npas4 mRNA expression in a small subpopulation. Indeed, cocaine conditioning significantly increased overall Npas4 mRNA expression in the MSN_Drd1+_1, MSN_Drd1+_2, MSN_Drd2+_1, and MSN_Grm8+_1 cell clusters (Fig. 2G). To determine the cell types of "highly expressing" Npas4+ neurons, we examined the top quartile of Npas4 RNA expression in the treatment groups. We observed Npas4 RNA expression above this threshold in approximately 2% of neurons from the cocaine conditioning group compared to only 1% of neurons from the home-cage control group (Fig. 2F). The largest Npas4+ cell types induced by cocaine conditioning was the MSN clusters (i.e., 47% D1-MSNs, 25% D2-MSNs) (Fig. 2F). Given that non-canonical Grm8-MSNs comprise ~2–6% of Npas4+ cells, we confirmed the expression of Npas4 mRNA in Grm8-expressing MSNs of the NAc using RNAscope (Fig. S2F). Finally, to investigate potential unique transcriptomic features of neurons recruited to the Npas4+ ensemble during cocaine conditioning, we conducted an additional analysis of differentially expressed genes (DEGs) in Npas4 mRNA-positive vs. -negative nuclei (Supplementary Data 2). We found 1463 significant DEGs across all cell types, including 206 and 254 DEGs in the D1- and D2-MSN clusters, respectively. Functional enrichment analysis of these clusters revealed DEGs linked to learning, memory, and synapse organization, among others (Fig. S2G, H), suggesting possible unique features of the cell populations that express Npas4 mRNA. Interactive snRNA-seq data from this experiment is available at https://bioinformatics-musc.shinyapps.io/Jessica_NAc_Cocaine_Npas4/.

## NPAS4 functions in D2R-expressing neurons to promote cocaine-seeking behaviors

To test possible cell type-specific role(s) for NPAS4 in NAc D1R- and D2R-positive neurons, we generated and validated a neurotropic, Cre-dependent Npas4 shRNA virus (AAV2-SICO-shNpas4) in male and female Drd1a-Cre or Drd2-Cre mice[52,53] (Figs. 3A and S3A–F). Following bilateral intra-NAc viral injections (Figs. 3A and S3G), we tested the mice sequentially in the cocaine CPP and a two-injection cocaine locomotor sensitization test (see methods for details) (Fig. 3B). Selective reduction of NPAS4 in NAc D1R-expressing neurons had no significant effect on cocaine CPP (Figs. 3C and S3H), cocaine-induced locomotion, or cocaine locomotor sensitization (Fig. S3J). In contrast, reduction of NPAS4 in D2R-expressing neurons completely abolished cocaine CPP (Figs. 3D and S3I), and it produced a very modest, but statistically significant, reduction of cocaine-induced locomotion (i.e., main effect of virus on cocaine injection days) (Fig. S3K). To increase our confidence that NPAS4 is required in NAc D2R+ neurons for cocaine CPP, and not due to off-target effects of the shRNA, we designed and validated an additional, sequence-independent shRNA (shNpas4-2) (Fig. S3L, M) and observed a significant reduction of cocaine CPP in the Drd2-Cre mice (Fig. S3N, O).

We next employed the rat intravenous cocaine self-administration (SA) assay, a robust SUD model that incorporates volitional drug use and relapse-associated behaviors, such as cue-reinstated drug seeking[54]. Using NAc-validated D1-Cre or D2-Cre BAC transgenic rats[55], we infused Cre-dependent shNpas4 or shScram viruses into the NAc core (NAcore)—a NAc subregion required for cue-reinstated cocaine seeking (Fig. S4I)[4,5]. Following 2 weeks of daily, 2-h cocaine SA sessions in D1- or D2-Cre rats (Fig. 3E), we observed no significant differences produced by shNpas4NAcore on cocaine SA acquisition, number of drug infusions, operant discrimination of the active vs. inactive lever, or extinction of responding on the lever formerly paired with cocaine

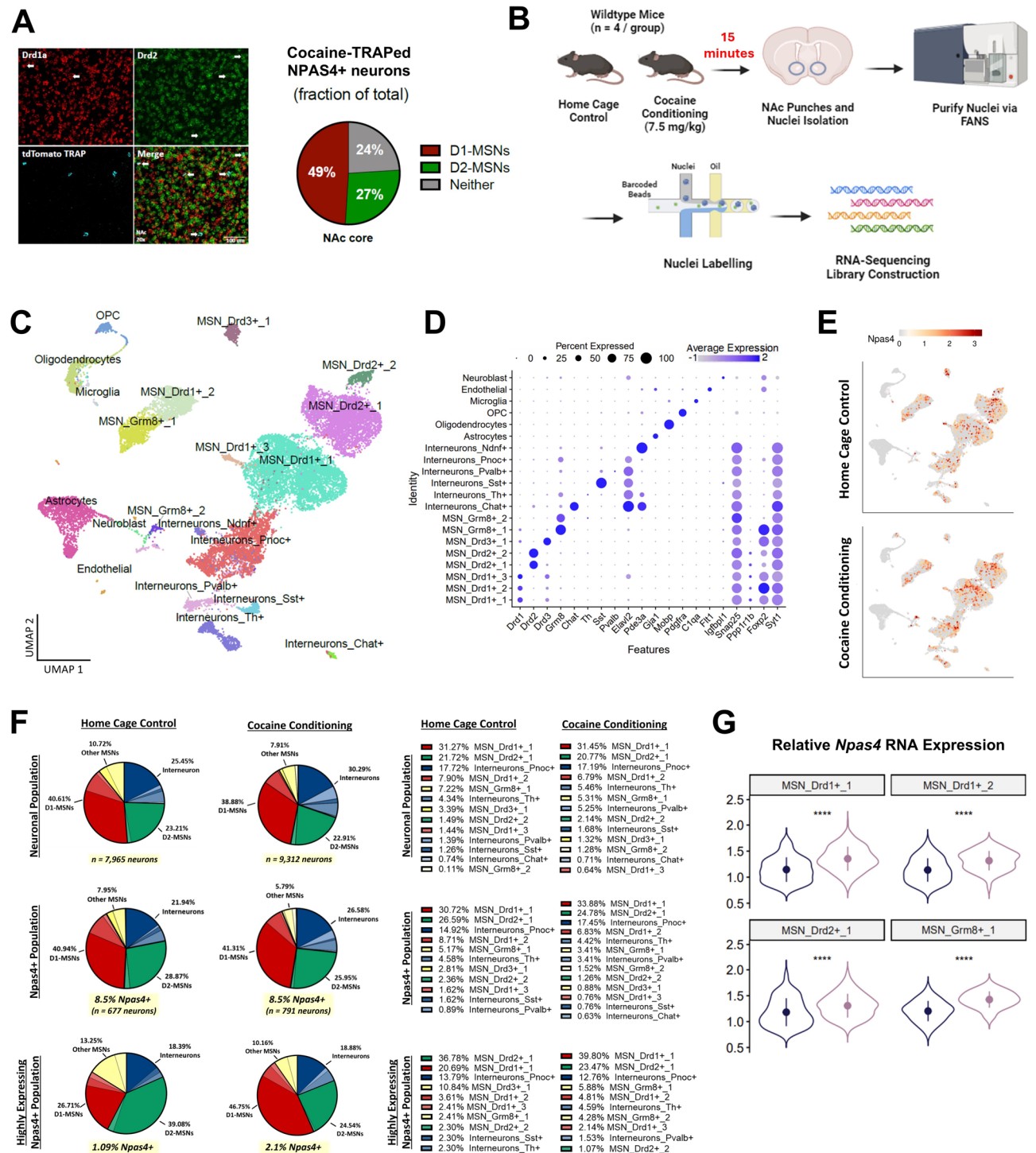

**Fig. 2 | NPAS4 is induced by cocaine conditioning predominantly in MSNs.**
**A** Representative RNAscope images (left) with probes for Drd1a, Drd2, and tdTomato in NAc slices of NPAS4-TRAP/Ai14 mice that received 4OHT after two cocaine conditioning sessions and quantification (right) of co-localized tdTomato (*Npas4*) expression with Drd1a or Drd2 markers (*n* = 11 mice). **B** Experimental design for snRNA-seq (*n* = 4 mice/group). **C** Uniform manifold approximation and projection (UMAP) plot of NAc single cells colored by cell type. Cell types were defined using known markers and confirmed by predictive modeling using a single-cell NAc atlas (see "Methods" and Fig. S2). **D** Dot plot showing the cluster-specific markers used to generate UMAP. Gray, low gene expression; dark blue, high gene expression. The

size of the circle represents the percentage of cells expressing highlighted genes. **E** Nebulosa plots depicting *Npas4* expression in the home cage and cocaine conditioning experimental groups. **F** Pie charts showing the cell type distribution of neuronal populations (top), *Npas4*+ populations (middle), and the highest *Npas4*-expressing populations. **G** *Npas4* expression is significantly increased for Drd1+_1, Drd1+_2, Drd2+_1, and Grm8+_1 cell types after cocaine conditioning (nuclei collected from *n* = 4 mice/group; Wilcoxen rank-sum test). Data are shown as mean ± SEM; ****p < 0.0001. Source data are provided in the Source Data file (**A**, **F**) or see "Data availability" for source data (**C**–**E**, **G**).

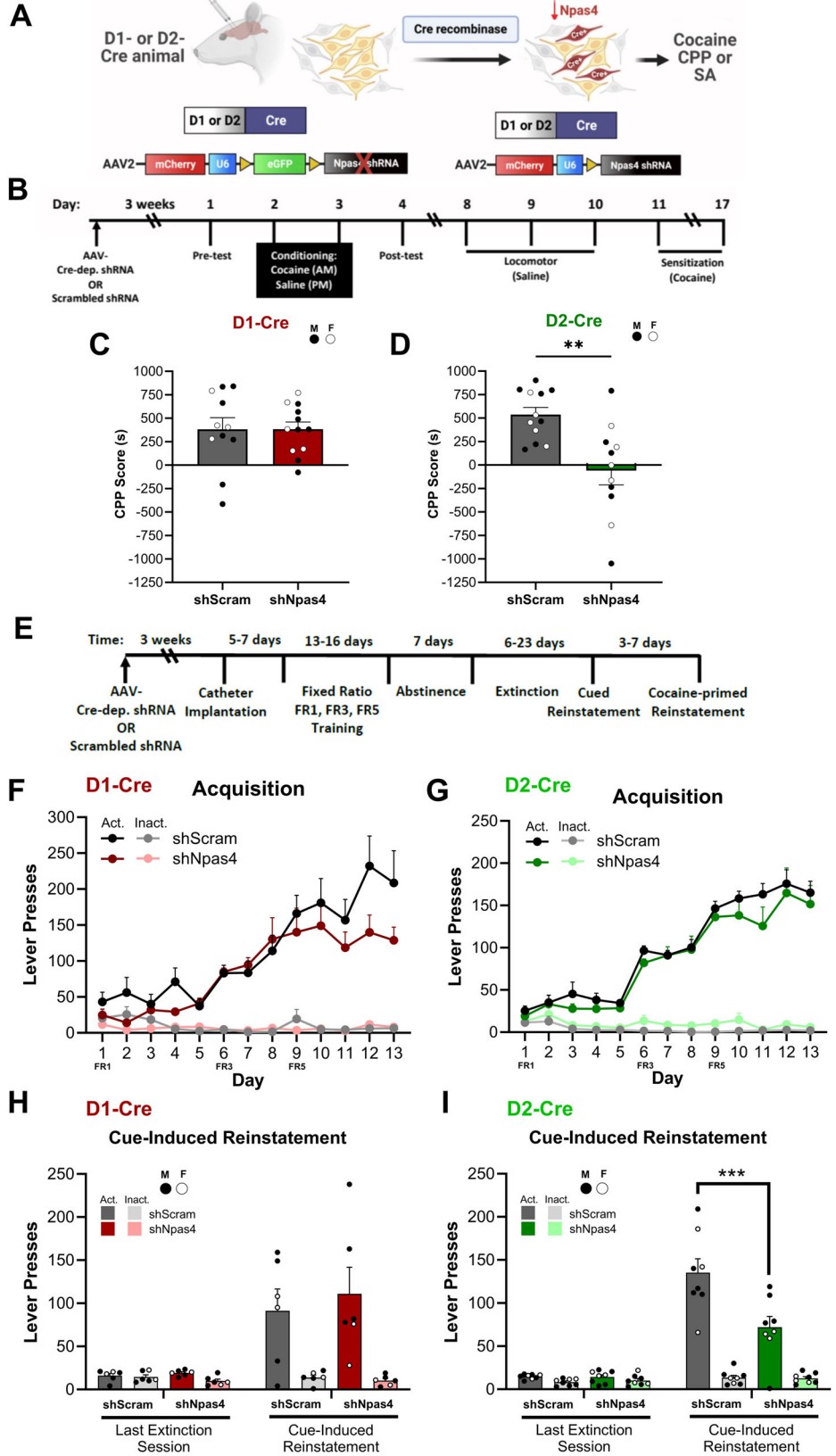

**Fig. 3 | NPAS4 functions in NAc D2-MSNs to promote cocaine-seeking behaviors. A** Experimental design using Cre-dependent *Npas4* shRNA in D1- or D2-Cre mice (CPP) and rats (IVSA). **B** Timeline of cocaine CPP and locomotor sensitization. **C** Cocaine CPP in D1-Cre ($n$ = 11, 12 mice/group) and (**D**) D2-Cre mice ($n$ = 11, 12 mice/group; unpaired *t*-test, $t$ = 3.578, df = 21, $p$ = 0.0018). **E** Timeline of cocaine SA. **F** Lever presses during acquisition of cocaine SA following NPAS4 knockdown in D1-

Cre ($n$ = 6 rats/group) and (**G**) D2-Cre rats ($n$ = 8 rats/group). **H** Cue-induced reinstatement of cocaine seeking in D1-Cre ($n$ = 6 rats/group) and (**I**) D2-Cre rats ($n$ = 8 rats/group; two-way RM ANOVA with multiple comparisons, $p$ = 0.0003). Data shown are mean ± SEM; **$p$ < 0.01, ***$p$ < 0.001. Source data are provided in the Source Data file.

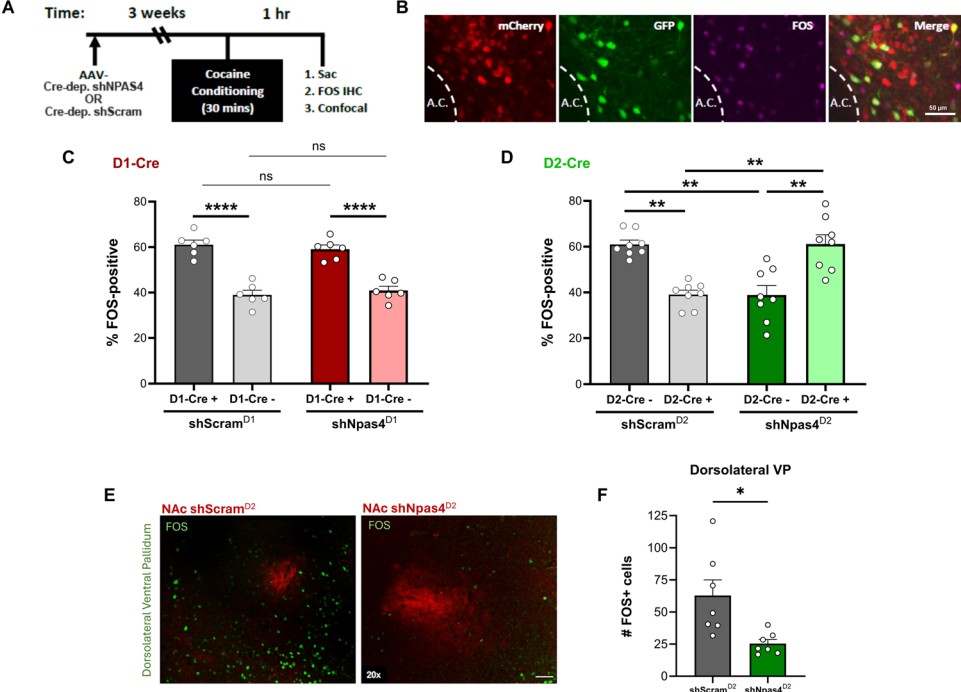

**Fig. 4 | NPAS4 regulates the preponderant activation of NAc D1-MSNs following cocaine conditioning. A** Timeline for cocaine CPP and FOS IHC. **B** Representative FOS IHC images showing Cre-dependent *Npas4* shRNA in the NAc of D2-Cre mice. **C** Quantification of FOS expression in D1- (*n* = 6 mice/group; two-way ANOVA, sig. main effect of cell type, *f* = 105.7, df = 1,20, *p* = <0.0001) and (**D**) D2-Cre mice (*n* = 8 mice/group) after cocaine CPP, including effects on "putative" MSNs after cell type-specific NPAS4 knockdown (two-way ANOVA with multiple comparisons, df = 28,

*p* = 0.0002). **E** Representative image showing FOS expression in the dlVP surrounding mCherry-labeled fibers from the NAc. **F** Quantification of FOS expression in the dlVP after NPAS4 knockdown in NAc D2-MSNs (*n* = 8 and 7 mice/group; unpaired *t*-test, *t* = 2.973, df = 12, *p* = 0.0116). Data shown are mean ± SEM; *\**p* < 0.05, \*\**p* < 0.01, \*\*\**p* < 0.001, \*\*\*\**p* < 0.0001. Source data are provided in the Source Data file.

(Figs. 3F, G and S4A–C, E–G). We then presented the rats with the non-extinguished light and tone cues formerly linked with drug delivery and measured drug-seeking behavior (i.e., non-reinforced pressing of the drug-paired lever). While NAcore shNpas4[D1-Cre] appeared to have no effect on cue-reinstated drug seeking (Fig. 3H), NAcore shNpas4[D2-Cre] significantly reduced cue-reinstated drug seeking (Fig. 3I). Of note, there was relatively high variability in cued seeking of the D1-Cre rats, potentially precluding our ability to detect an effect. Indeed, in this dataset, we observed an effect size (*f*) of 0.132 on the non-significant interaction of virus and session, and a power analysis determined that a total sample size of 132 animals would be required to achieve a power of β = 0.85. Importantly, however, in the D2-Cre rats we observed an effect size of 0.85 and achieved a power of β = 0.99. Also of note, NAcore shNpas4 had no effect on cocaine-primed reinstatement in either D1- or D2-Cre rats (Fig. S4D, H), suggesting that NAcore NPAS4 plays a selective role in external drug-cue associations. Next, we tested whether NPAS4's NAc D2-MSN role in cue-reinstated cocaine seeking was due to general reward-related cue associations. Using a similar operant sucrose SA design in D2-Cre rats (Fig. S4J), we observed no effects of NAcore shNpas4[D2-Cre] on the acquisition or extinction of sucrose SA (Fig. S4K, N), but in contrast to reduced cue-reinstated cocaine seeking, we observed a significant increase in cue-reinstated seeking of sucrose (Fig. S4O). NAcore shNpas4[D2-Cre] had no effect on sucrose-primed seeking behavior (Fig. S4P). Taken together, our data reveal essential roles for NPAS4 in NAc D2-MSNs in both contingent and non-contingent models of drug seeking and opposing roles for cue-induced seeking of cocaine versus sucrose.

## NPAS4 regulates the preponderant activation of NAc D1-MSNs following cocaine conditioning
Since activation of NAc D2-MSNs reduces drug-seeking behaviors[7,14–16,18–23,25,56–59] and loss of NPAS4 in NAc D2-expressing

neurons reduced both cocaine CPP and cue-reinstated cocaine seeking, we next assessed the cell type-specific effects of NPAS4 knockdown on the activation balance of NAc D1-MSNs and D2-MSNs after cocaine conditioning (Fig. 4A) using the activity-regulated IEG, FOS, as an indicator of recent neural activity. We injected Cre-dependent shScram or shNpas4 into the NAc of D1- and D2-Cre mice and observed that ~60% of FOS+ NAc cells in control shScram animals were either D1-positive (Fig. 4C, left) or D2-negative (Fig. 4D, left), while ~40% of FOS+ neurons were either D2-positive (Fig. 4D, left) or D1-negative (Fig. 4C, left), as expected from prior literature[60,61]. While knockdown of NPAS4 in NAc D1-cells had no effect on the D1:D2 FOS+ ratio (Fig. 4C), NPAS4 knockdown in NAc D2-cells produced a significant shift in the D1:D2 FOS+ ratio (Fig. 4D), characterized by a significant increase in the proportion of FOS+ D2-positive cells and a significant reduction in the proportion of FOS+ D2-negative cells. In the D2-Cre mice with NPAS4 knockdown, we observed no difference in the total number of cocaine conditioning-induced FOS+ neurons compared to shScram controls (Fig. S5A), but there was a significant interaction between virus and cell-type for the number of FOS+ cells (Fig. S5B), with a statistical trend toward an increase in FOS+ D2-positive cells in the shNpas4 condition (Fig. 4D). Consistent with this observation of a D1:D2 activation shift, after cocaine conditioning we observed a significant reduction in FOS+ neurons in the dorsolateral ventral pallidum (dlVP) (Fig. 4E, F), a key projection target of NAc D2-MSNs that is required for cued drug seeking[14,18,24,25,62–64].

## NPAS4 regulates the cocaine conditioning transcriptome in NAc D2-MSNs
To investigate the gene expression programs regulated by both cocaine conditioning and NPAS4, we performed snRNA-seq of NAc tissue isolated from wild-type mice expressing either AAV2-shNpas4 or AAV2-shScram virus. Mice were subjected to either cocaine

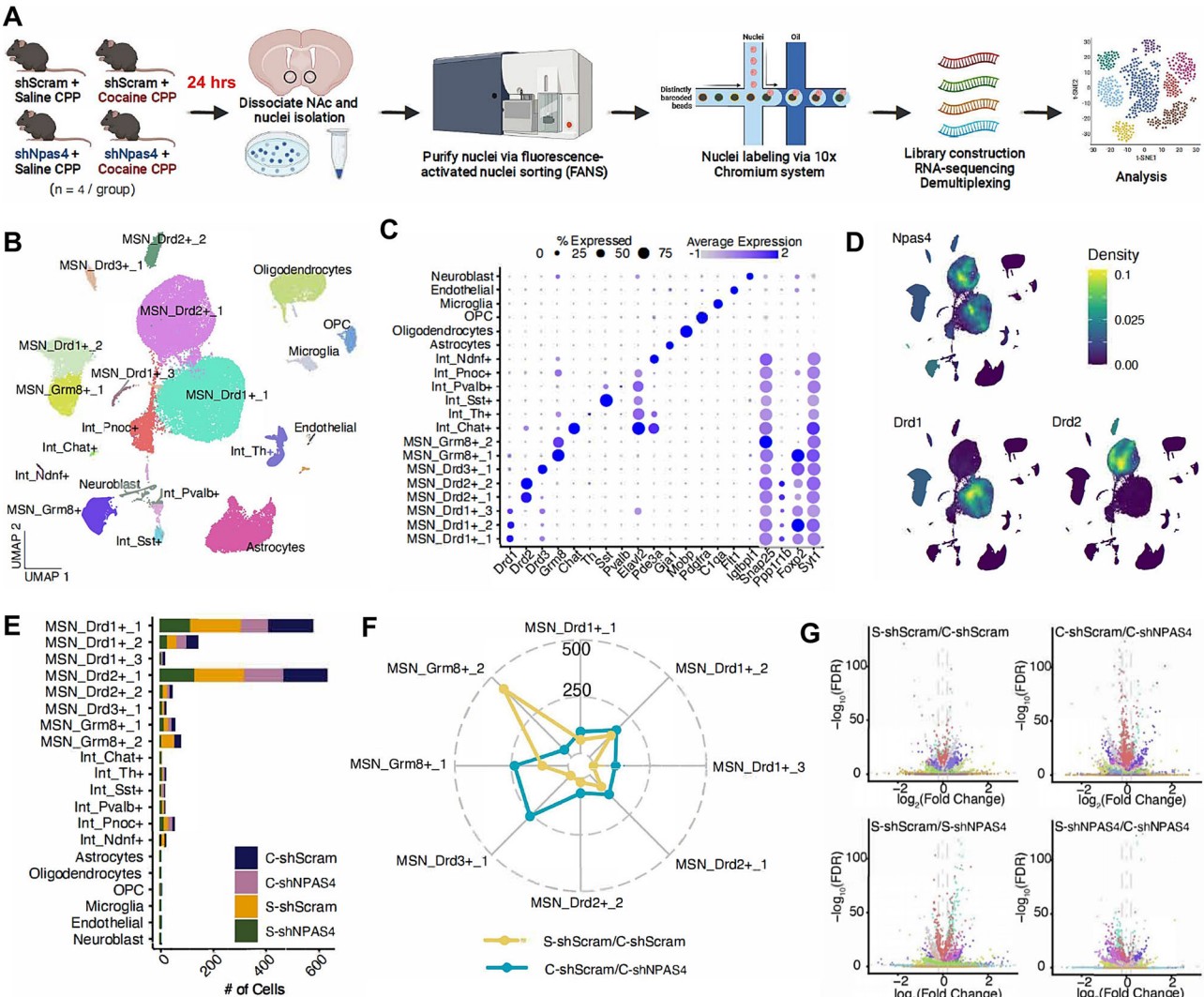

**Fig. 5 | NPAS4 regulates the cocaine-conditioned NAc transcriptome.**
**A** Experimental design for snRNA-seq ($n = 4$ mice/group). **B** Uniform manifold approximation and projection (UMAP) plot of NAc single cells colored by cell type. Cell types were defined using known markers and were confirmed by predictive modeling using a single-cell NAc atlas. **C** Dot plot showing the cluster-specific markers used to generate UMAP. Gray, low gene expression; dark blue, high gene expression. The size of the circle represents the percentage of cells expressing highlighted genes. **D** Nebulosa plots depicting NPAS4 expression in D1-, D2-, and Grm8-MSNs. **E** Number of NPAS4 cells in each cluster, colored by experimental replicate. **F** Radar plot showing the number of differentially expressed genes in each identified MSN cell type. The comparison between saline and cocaine CPP shScram control groups is shown in yellow, and the comparison between cocaine CPP shScram and shNpas4 is shown in blue. DEGs are defined by FDR < 0.05 and abs(log2(FC)) > 0.2. **G** Volcano plots highlighting the transcriptomic differences in each comparison, colored according to cell type-specific clusters. *Y*-axis corresponds to the log10(FDR) and *x*-axis corresponds to the log2(Fold Change). See "Data availability" for source data.

(7.5 mg/kg; i.p.) or saline-only CPP, and 24-h after the last conditioning session, NAc tissues were rapidly extracted and processed for snRNA-seq (Fig. 5A). As NPAS4 induction is very rapid and transient, we chose to examine the 24-h time point to detect more stable, downstream NPAS4-dependent DEGs in D2-MSNs that are associated with the reduced expression of cocaine CPP behavior. Using the same, previously described approach (Fig. S5C–F), we first defined and analyzed the NAc cell-type clusters (Fig. 5B, C and Supplementary Data 3). Similar to our other snRNA-seq experiments (Fig. 2), we found that *Npas4* mRNA expression was detected almost exclusively in neurons, with the vast majority (~85%) found in MSN clusters, including MSN_Drd2+, MSN_Drd1+, and MSN_Grm8+ (Fig. 5D, E). Interestingly, cocaine conditioning produced DEGs in MSNs (Fig. 5F), interneurons, and non-neuronal cells, including astrocytes and oligodendrocytes (Fig. S5G), and neuronal *Npas4* mRNA knockdown, combined with cocaine conditioning, increased the number of MSN and non-neuronal DEGs (Fig. 5G), the latter

presumably due to cell non-autonomous effects of neuronal NPAS4 knockdown (Fig. S5G).

To determine cell type-specific transcriptomic influences of NAc NPAS4 that might be relevant to its role in facilitating drug-related behavior, we next conducted a targeted analysis of DEGs across all four experimental conditions specifically in the D2-MSN cell cluster. We detected 332 DEGs (Figs. 5F and 6A, D) in the *Drd2*-positive cell cluster following cocaine conditioning, with the vast majority of the DEGs significantly upregulated with *Npas4* knockdown. Interestingly, at the 24-h post-conditioning timepoint, we detected only 24 unique genes that were significantly altered in the D2-MSN cluster (Fig. 6D). Analysis of DEGs from the cocaine conditioning animals revealed enrichment for genes linked to cocaine, amphetamine, brain development, and synapses (Fig. 6B, top). Under saline-only CPP conditions, we observed that shNpas4 altered the expression of 334 genes, including *Penk*, *Calm1*, *Cartpt*, and *Camkv*, and *Cacna1h* (Fig. 6A), suggesting an important role for NPAS4 in the cocaine-independent D2-MSN

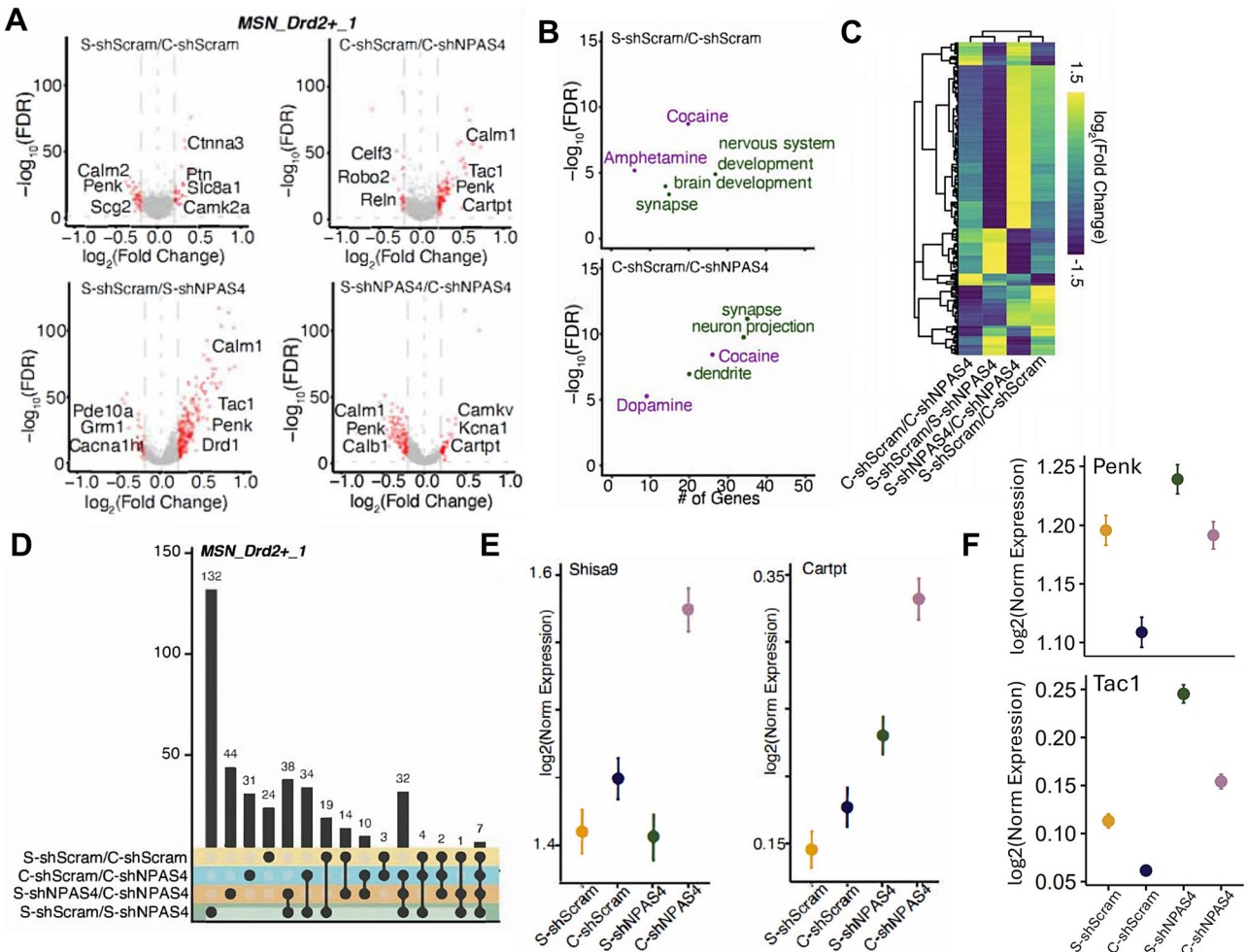

**Fig. 6 | NPAS4-mediated differential gene expression in NAc D2-MSNs.**
**A** MSN_Drd2+-specific volcano plots showing upregulated and downregulated genes (DEGs) in each group comparison. $Y$-axis corresponds to the $-\log_{10}(FDR)$ whereas $x$-axis corresponds to the log2(Fold Change). DEGs are defined by FDR < 0.05 and abs(log$_2$(FC)) > 0.2. Genes of interest are highlighted. **B** Scatter plots depicting functional enrichment of S-shScram vs. C-shScram (top) and C-shScram vs. C-shNPAS4 (bottom). $Y$-axis correspond to the $-\log_{10}(FDR)$ whereas the $x$-axis correspond to the number of genes within the categories. In purple, categories related to drugs. Green: functional categories related to molecular functions. Analysis based on the Fisher's exact test. **C** Heat map showing fold-change similarities and differences between the four comparisons in D2-MSNs. Hierarchical clustering based on Pearson's correlation. **D** Upset plot showing overlapping DEGs between groups in D2-MSNs. The bar graph at the top indicates the number of intersections. **E** Error plot showing the median expression of DEGs in D2-MSNs significantly upregulated ONLY in the "Cocaine CPP + shNPAS4" group, Shisa9 (left) and Cartpt (right), as well as (**F**) genes specifically downregulated by cocaine and upregulated by shNpas4, Penk (top), and Tac1 (bottom) (nuclei collected from $n$ = 4 mice/group). $X$-axis correspond to the experimental categories whereas the $Y$-axis the scaled expression level. Error bars correspond to the standard error of the mean. See "Data availability" for source data.

transcriptome. In the cocaine CPP-treated condition, shNpas4 produced many DEGs, such as *Calm1*, *Cartpt*, *Reln*, *Penk*, and *Robo2* (Fig. 6A and Supplementary Data 4), with reported functions related to cocaine, amphetamine, synapses, neuronal projections, dendrites, and dopamine (Fig. 6B, bottom). Of the DEGS regulated by both cocaine and NPAS4, a few genes were of particular interest, including *Shisa9* and *Cartpt* (Fig. 6E). *Shisa9* is involved in regulating neuronal plasticity[65], while *Cartpt* (cocaine and amphetamine-regulated transcript prepropetide) overexpression negatively regulates cocaine CPP[66]. We also investigated genes downregulated by cocaine, but upregulated by shNpas4, such as *Penk* and *Tac1* (Fig. 6F). When we analyzed DEGs in the major D1-MSN cluster (Fig. S6A), we found that shNpas4 had a smaller impact on D1-MSN gene expression compared to D2-MSNs (i.e., 195 in the saline condition and 96 in the cocaine condition; Fig. S6D), and functional analysis of D1-MSN DEGs revealed enrichment for genes linked to cocaine, synapse organization and signaling, neuronal development, and dopamine-regulated processes (Fig. S6B). A comparison of shNpas4 DEGs in D1- and D2-MSNs revealed non-overlapping DEGs (Fig. S6D); however, a majority of the DEGs

were expressed in both cell types, and most of the overlapping DEGs showed a similar magnitude and direction of change (Fig. S6E). Interactive snRNA-seq data from this experiment is available at https://bioinformatics-musc.shinyapps.io/Hughes_NAc_Cocaine_NPAS4/.

## NPAS4 blocks cocaine conditioning-induced dendritic spine growth and strengthening of prelimbic cortical inputs onto NAc D2-MSNs

Since reduction of NPAS4 in NAc D2-MSNs[1]: reduces cocaine CPP and cue-reinstated cocaine seeking[2], inverts the balance of D1:D2-MSN activation during cocaine conditioning[3], reduces activation of an essential downstream brain region (dlVP) required for drug seeking, and[4] significantly alters the expression of genes linked to synapses, dendrites, neuron projections, and cocaine, we next examined NPAS4's role in NAc D2-MSNs on glutamatergic dendritic spines, the location of most excitatory synapses on MSNs. Using a Cre-dependent viral approach in D2-Cre mice, we found with planned post hoc comparisons that NPAS4 knockdown in mice conditioned with cocaine, but not saline (Fig. 7A), increased D2-MSN dendritic spine density

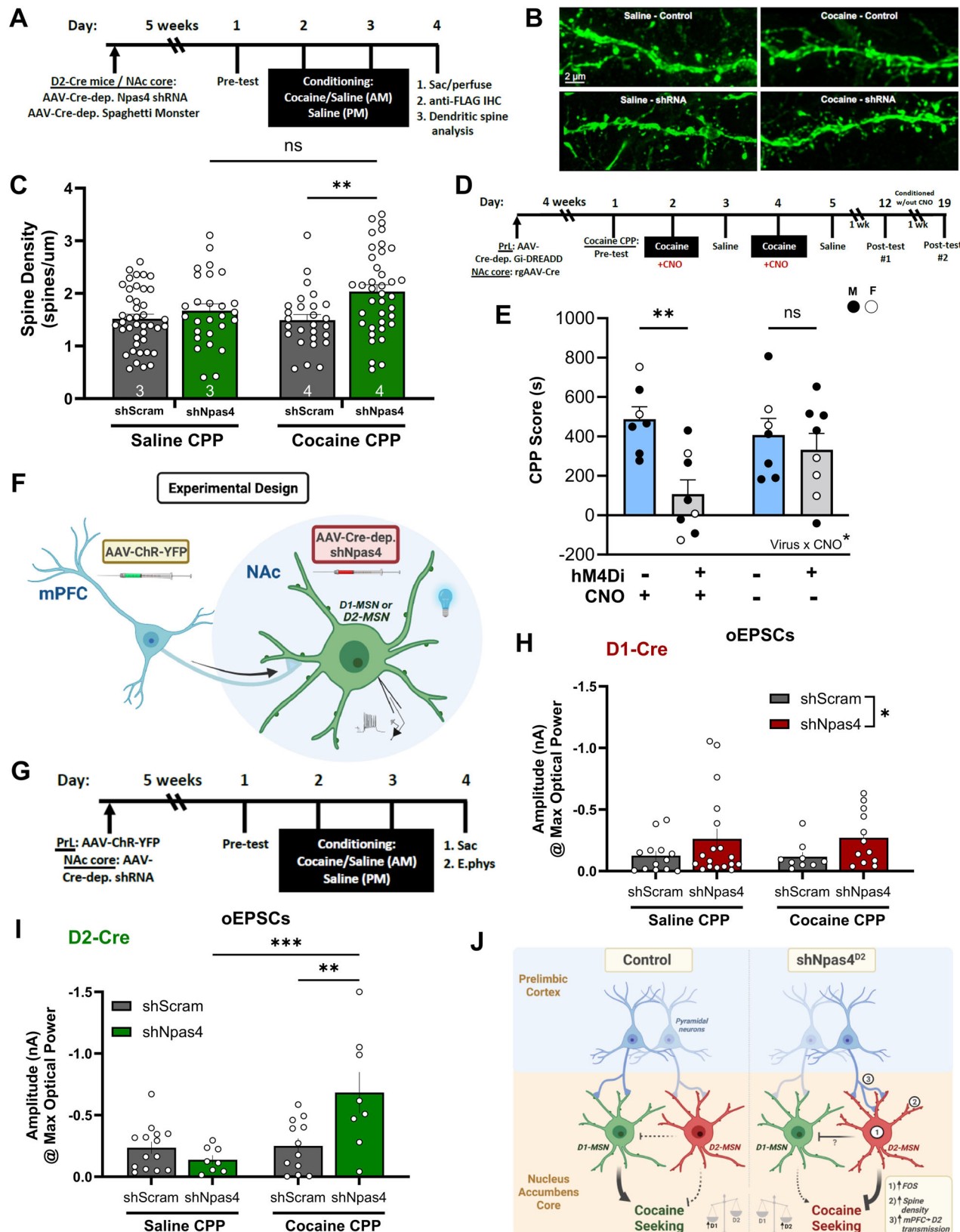

(Fig. 7B, C). This increase in density was observed mostly in spines with a head diameter <0.5 mm (Fig. S7A), suggesting that NPAS4 blocks the formation of new "thin"-type spines, which are highly motile and plastic[67,68]. Interestingly, in saline-treated mice, no D2-MSN spine changes were observed with shNpas4, indicating a specific interaction between NPAS4 and cocaine conditioning.

We measured the effects of shNpas4 and cocaine conditioning on functional glutamatergic synapses on D1- and D2-MSNs by measuring spontaneous excitatory postsynaptic currents (sEPSCs), which assesses the combined excitatory synapses onto MSNs (see "Methods"; Fig. S7D–G). Cocaine conditioning and/or shNpas4 in D1-MSNs produced no significant changes in sEPSC frequency or amplitude

**Fig. 7 | NPAS4 blocks cocaine conditioning-induced dendritic spine growth and strengthening of prelimbic cortical inputs onto NAc D2-MSNs. A** Timeline for behavior and dendritic spine labeling. **B** Representative images of dendritic spines on NAc D2-MSNs. **C** Quantification of spine density on D2-MSNs after cocaine ($n = 27$ and 36 dendritic segments/group from 4 mice/group, $p = 0.0068$) and saline CPP ($n = 42$ and 27 dendritic segments/group from 3 mice/group). **D** Timeline for cocaine CPP experiment inhibiting the PrL-NAc core circuit and (**E**) behavioral results following cocaine CPP ($n = 7$, 8 mice/group; two-way ANOVA with multiple comparisons, df = 26, $p = 0.0035$). **F** Experimental design for stimulating mPFC

afferents to NAc MSNs. **G** Timeline of cocaine CPP before electrophysiology. **H** Evoked EPSCs onto D1-MSNs and (**I**) D2-MSNs from the mPFC after NPAS4 knockdown in the corresponding cell type (two-way ANOVA with multiple comparisons; D1-Cre mice: shScram $n = 13$ and 9 cells/group, shNpas4 $n = 18$ and 12 cells/group; D2-Cre mice: shScram $n = 14$ and 12 cells/group, shNpas4 $n = 8$ cells/group, $p = 0.0009$, 0.0046). **J** Graphical illustration showing the effects of NAc shNpas4$^{D2}$ on MSN activity, dendritic spine density, excitatory transmission from the mPFC, and subsequent cocaine-seeking. Data shown are mean ± SEM; *$p < 0.05$, **$p < 0.01$, ***$p < 0.001$. Source data are provided in the Source Data file.

(Fig. S7D, F). Similar to previous studies on the effects of abused substances on D2-MSNs[15,58,69,70], we observed that cocaine conditioning reduced sEPSC frequency (Fig. S7E), but not amplitude (Fig. S7G), in control D2-MSNs, but the sEPSC frequency reduction was abolished by shNpas4 (Fig. S7E). These data suggest that NPAS4 plays an important role in regulating drug-induced changes in excitatory synaptic transmission onto D2-MSNs.

One of the strongest glutamatergic inputs onto NAc D2-MSNs comes from the mPFC[71,72], and prelimbic (PrL) mPFC pyramidal neuron projections to the NAcore are required for cue-reinstated cocaine seeking in rats[1,9,11,12,73–75]. To test the importance of the PrL → NAcore circuit for mouse cocaine CPP, we infused a retrograde virus expressing Cre-recombinase (AAVrg-Cre) in the NAc, and a Cre-dependent Gi-DREADD in the mPFC (AAV2-hM4Di-mCherry) to allow for CNO-dependent suppression of PrL → NAcore projection activity (Figs. 7D and S7B, C). We observed that Gi-DREADD-mediated suppression of the PrL → NAcore circuit during cocaine conditioning significantly reduced expression of cocaine CPP (Fig. 7E, left). Subsequent rounds of cocaine conditioning, in the absence of CNO, allowed for the development of normal place preference behavior in the same animals that previously received CNO administration (Fig. 7E, right).

Since the PrL → NAcore circuit is important for both cocaine CPP and cue-reinstated cocaine seeking, we examined PrL-specific synaptic transmission onto NAc D1- or D2-MSNs. To this end, we combined NAc-specific virus injections of Cre-dependent shNpas4 in D1-Cre or D2-Cre mice with PrL-targeted virus injections of channelrhodopsin (AAV-hSyn-hChR2(H134R)-eYFP) and subjected them to our standard 2-day cocaine or saline conditioning procedure (Fig. 7F, G). After 24-h, we generated and analyzed coronal, acute slice recordings of blue light-evoked excitatory postsynaptic currents (oEPSCs) in NAc D1- or D2-MSNs (Fig. 7H, I). shNpas4 in D1-MSNs produced a small, but significant, increase in PrL → NAcore oEPSC amplitude that was independent of cocaine conditioning (i.e., main effect of shNpas4; no interaction) (Fig. 7H). In contrast, reduction of NPAS4 in D2-MSNs produced a significant interaction between cocaine conditioning and shNpas4 virus, compared to the scrambled shRNA control (Fig. 7I). Post hoc analyses revealed a robust increase in PrL → D2-MSN$^{NAcore}$ glutamatergic synaptic transmission in animals that received both shNpas4 virus and cocaine conditioning (Fig. 7I). These data revealed that NPAS4 in NAc D2-MSNs functions to limit strengthening of PrL → NAcore D2-MSN inputs produced by cocaine conditioning, ultimately facilitating the D1-MSN:D2-MSN activation imbalance critical for drug seeking (Fig. 7J).

## Discussion

A fundamental function of the brain is to form enduring associations between salient experiences and environmental cues, and neuronal activity-dependent mechanisms are important for this process[61,76]. The molecular underpinnings of the link between environmental cues and reward associations are crucial in SUD, as powerful drug-cue associations often trigger drug seeking and relapse[77]. The mechanisms by which drug-conditioned cues produce and maintain the preponderant activation of D1-MSNs to support cocaine-seeking behavior remain unclear. Here we show that NPAS4 plays an essential, cell type-specific role in the NAc to facilitate drug-seeking behavior, likely by regulating

the activation balance of NAc D1- and D2-MSNs. Using a new NPAS4-TRAp2a mouse combined with chemogenetics, we found that an NPAS4-expressing subpopulation of NAc neurons is required for expression of cocaine CPP. Single-nucleus transcriptomic analyses and in situ approaches from cocaine-conditioned mice revealed that *Npas4* is expressed predominantly in NAc MSNs, and despite its induction in more D1- than D2-MSNs, we found that NPAS4 in NAc D2-MSNs, but not D1-MSNs, is required for cocaine-context associations and cued reinstatement of cocaine-, but not sucrose-, seeking behavior. We also show evidence that NPAS4 in NAc D2-MSNs influences the activation balance of NAc D1- and D2-MSNs, as measured by FOS induction, and that this correlates with NPAS4's ability to block cocaine conditioning-induced increases in dendritic spine density and PFC excitatory synaptic transmission onto NAc D2-MSNs. Analysis of differential gene expression in NAc D2-MSNs revealed that NPAS4 and cocaine conditioning regulate multiple genes whose functions are linked to synapses, neuronal projections, and the cocaine-regulated transcriptome, positioning NPAS4 as a critical negative regulator of cocaine-induced D2-MSN synaptic plasticity. Together, our data demonstrate that NAc NPAS4-expressing neurons form a functional ensemble necessary for cocaine-conditioned cue memories, and that NPAS4 functions in NAc D2-MSNs to block cocaine-induced adaptations that oppose cocaine-conditioned cue/context associations.

The activity of NAc D1- and D2-MSNs influences several drug-related behaviors, with activation of D1-MSNs typically promoting, and D2-MSNs typically opposing, drug behaviors[13,15,21,59]. Analogous to the drug-induced expression of FOS[60,78], we found that cocaine conditioning induced more Npas4+ NAc D1-MSNs than NAc D2-MSNs. The preponderant D1-MSN activation is likely produced by the well-studied influences of dopamine via D1-like receptor activation and PKA activation on glutamatergic synaptic transmission, the activity-dampening effects of D2-like receptor activation, and the known glutamatergic input differences onto these two NAc cell populations[15,72,79]. Our two independent snRNA-sequencing experiments revealed the preponderant expression of NPAS4 in D1- and D2-MSN cell clusters (>65%). At 15 min post-cocaine conditioning, which is near the peak of *Npas4* mRNA expression, we detected a significant increase in *Npas4* mRNA, with the vast majority (~75%) of *Npas4*+ neurons grouped into the D1- and D2-class MSN clusters. Moreover, our RNAscope in situ hybridization analysis using the NPAS4-TRAp2a mouse revealed that ~75% of cocaine-conditioned TRAP cells in the NAc are D1- or D2-positive (~50% D1 and ~25% D2). The non-D1- or D2-MSN *Npas4*+ mRNA populations were a mixture of non-canonical Grm8+ MSNs and several interneuron populations, but not ChAT+ interneurons. Notably, the *Npas4* mRNA-positive cells (~8.5% of NAc neurons) in either home-caged (basal) or cocaine-conditioned mice appear to be distributed proportionally across most NAc neuronal populations. Following cocaine exposure, *Npas4* mRNA expression is significantly increased in Drd1+, Drd2+, and Grm8+ MSN cell clusters. Indeed, analysis of the highest quartile of *Npas4* mRNA expression is detected in approximately 2% of NAc neurons, and more than 80% of those neurons were MSN cell types (i.e., 47% in D1-MSNs, 25% in D2-MSNs, and 10% in Grm8+ MSNs), and these proportions were similar to those observed in the NPAS4-TRAp2a population (i.e., 49% in D1+, 27% in D2, and 24% non-D1/D2). Immunostaining of endogenous NPAS4 protein in *Drd1*-

tdTomato:*Drd2*-EGFP BAC transgenic mice at 60 min after cocaine conditioning revealed a similar distribution of D1:D2 (i.e., 60% D1R+, 40% D2R+), but nearly half of the NPAS4+ cells did not co-express tdTomato (D1R) or EGFP (D2R). There could be several explanations for this discrepancy, including: (1) the possibility that bacterial artificial chromosome (BAC) reporter transgenes do not fully recapitulate full endogenous *Drd1* and *Drd2* expression, since one or more copies of the reporter gene cassette are randomly inserted into the genome[80–82], (2) the possibility that protein translation is regulated differently than mRNA expression, and/or (3) the possibility that the D1/D2R-reporter gene expression in approximately 25% of the neurons falls below the fluorescence detection threshold. Indeed, we previously detected approximately 15% of the NPAS4+ cells co-expressed the MSN marker, DARPP-32[42], leading us to suspect that NPAS4 is induced by cocaine conditioning in mostly non-MSN neuronal populations. However, considering our current findings using next-generation sequencing and RNAscope, we suspect that the mouse monoclonal anti-DARPP-32 immunostaining underestimated the fraction of NPAS4+ MSNs in our previous study[42].

While NPAS4 is induced by cocaine conditioning predominantly in NAc D1- and D2-MSNs, we found that its function in NAc D2-MSNs is critical to support cocaine CPP and cued cocaine seeking. Interestingly, for cocaine CPP, reduction of NPAS4 in NAc D2-MSNs appeared to not only limit the development of place preference, but it also created an aversion to the cocaine-paired environment in many mice (Figs. 3D and S3O). This is particularly interesting given that we observed no differences in drug-taking behavior during the acquisition phases of cocaine self-administration. Despite the role of NPAS4 in NAc D2-, but not D1-, MSNs in mediating these behaviors, viral-mediated reduction of NPAS4 in D1-MSNs does significantly enhance PrL → NAcore D1-MSN glutamatergic synaptic transmission, but that effect was independent of cocaine conditioning. While NPAS4 knockdown in NAc D1-MSNs did not alter cocaine CPP or cue-reinstated cocaine seeking, we cannot rule out the possibility that D1-specific NPAS4 knockdown might enhance cocaine-conditioned behavior under sub-threshold cocaine dose conditions. Of note, a prior published study reported that Cre-dependent overexpression of a dominant-negative NPAS4 protein (NPAS4 DNA binding domain) in the NAc of *Drd1*-Cre mice reduced cocaine CPP, while no effects were observed in the NAc of *Adora2a*-Cre mice[83], which is opposite of our findings here. Comparing the studies, we noted differences in cocaine dose, the number of conditioning sessions, dominant-negative vs. shRNA approaches, Cre lines used, and validation of Cre-dependent manipulation of NPAS4; however, we cannot readily explain our difference in findings. To increase confidence in our results, we created a second unique shRNA sequence for Cre-dependent knockdown of *Npas4* mRNA and replicated our initial cocaine CPP findings in *Drd2*-Cre mice. Moreover, in our prior study[42], we showed that viral-mediated Cre expression in the NAc of *Npas4*-floxed mice recapitulated the cocaine CPP effects of our original AAV2-shNpas4.

We found that cell type-specific NPAS4 knockdown in NAc D2-MSNs (*Drd2*-Cre) produced a cocaine conditioning-dependent increase in their dendritic spine density; whereas, saline conditioning had no effect on D2-MSN spine density. However, a recent study reported that pan-neuronal NPAS4 knockdown in NAc caused a reduction in D1-MSN (i.e., AAV-PPTA-Cre), but not D2-MSN (i.e., AAV-ENK-Cre), dendritic spine density[43]. In our study, we specifically reduced *Npas4* expression in D2R+ neurons and assessed dendritic spine density in this cell type. As we did not detect any effects of *Npas4* manipulation in D1-MSNs on cocaine behavior, we did not examine D1-MSN dendritic spine density in this study.

One of the principal targets of NAc MSNs is the dlVP, and D2-MSNs make strong connections onto these neurons[24,64]. Optogenetic activation of NAc D2-MSNs that project to the dlVP attenuates contextual cocaine seeking[10,24,84]. Similar to this finding, the shift in the D1:D2-MSN

activation balance with NPAS4 knockdown in NAc D2-MSNs appears to have a profound consequence on the activation (i.e., FOS induction) of the dlVP, which is consistent with the reduction in cocaine seeking behavior. It is also interesting that NPAS4-dependent regulation of the PrL → NAcore D2-MSN circuit is important for place conditioning and drug-cue associations that promote cocaine seeking. Clearly, there are multiple glutamatergic inputs to the NAc that are important for regulating addiction-related behaviors in rodents, such as the ventral hippocampus (vHipp) and basolateral amygdala (BLA)[13,85], but PrL → NAcore excitatory transmission plays an essential role in cue-reinstated drug seeking[11,12], and we showed here that the activity of this same circuit is required for development of cocaine CPP, a previously unpublished result. Compared to NAc D1-MSNs, NAc D2-MSNs receive stronger excitatory input from the mPFC[72], suggesting the importance of NPAS4 expression in suppressing drug-induced strengthening of glutamatergic inputs onto D2-MSNs. While we focused here on the behavioral requirement of the mPFC → NAc circuit, it will be interesting in the future to explore the influence of NPAS4 on other major glutamatergic inputs onto NAc D2-MSNs, including those from the BLA and vHipp brain regions[13]. Similar to a prior report[58], the sEPSCs in D2-MSNs, which measure combined excitatory inputs from all sources, showed a significant decrease in frequency, but not amplitude, following cocaine conditioning. Reduction of NPAS4 in D2-MSNs blocks the cocaine conditioning-induced decrease in sEPSC frequency, which might reflect the net balance of a reduction of non-mPFC inputs and the increase in mPFC → NAcore D2-MSNs excitatory inputs, and/or an important role for NPAS4 in the reduction of non-mPFC synaptic inputs, which will be worth investigating in future studies.

There are likely multiple NPAS4 target genes (direct or indirect) in D2-MSNs that block cocaine conditioning-induced dendritic spine formation and strengthening of mPFC synaptic inputs. We noted several interesting upregulated DEGs, including *Shisa9* and *Cartpt*. Shisa9, initially named CKAMP44, is involved in regulating short-term neuronal plasticity by interacting with AMPA receptors on dendritic spines to promote excitatory synaptic AMPA receptor desensitization[65]. The Cartpt gene, encoding for a psychostimulant-regulated neuropeptide, CART, has previously been shown to limit cocaine CPP[66]. Another interesting DEG is *Penk*, a gene that encodes the reward-related neuropeptide, enkephalin, which binds to delta and mu opioid receptors with high affinity and is thought to limit GABA release by activation of DOR and MOR autoreceptor Gi-coupled signaling[86,87]. In the NAc, DORs are expressed postsynaptically on D2-MSNs[88] and on the terminals of PFC afferents[89]. In control mice, cocaine conditioning significantly reduced *Penk* expression in D2-MSNs, which was accompanied by a decrease in sEPSC frequency. However, the reduction in *Penk* mRNA was absent in cocaine-conditioned mice expressing NAc shNpas4, revealing a role of NPAS4 in *Penk* expression in NAc D2-MSNs. Calmodulin genes are also differentially-expressed following cocaine conditioning (e.g., downregulated *Calm2* in S-shScram vs. C-shScram) and shNpas4 (upregulated *Calm1* in C-shScram vs. C-shNpas4), which are involved in brain calcium signaling and differentially expressed in mice following cocaine SA[90]. Interestingly, in humans who abused cocaine, there is a reduction in calmodulin-related gene transcription[91]. It's worth noting that *Npas4* mRNA and protein are very short-lived (~60–90 min[34,92]). Therefore, we speculate that some of the D2-MSN DEGs at 24-h post-cocaine conditioning are indirect targets of NPAS4, but this must be tested empirically. We chose to analyze this time point because it corresponds to the precise time at which behavioral and synaptic changes are observed. A large fraction of the shNpas4 D2-specific DEGs were upregulated, and since NPAS4 typically increases expression of its target genes, many of these upregulated DEGs could be transcribed by secondary waves of gene expression produced by the interaction of shNpas4, cocaine conditioning, and drug-induced neuroadaptations. However, it is possible that NPAS4

functions as, or regulates, a transcriptional repressor, or that other IEGs outcompete Npas4 binding to the promoter regions of downstream genes at this time point, such as *Nr4a1*[66]. Future studies examining multiple time points after cocaine conditioning will be important for understanding the cascade of transcriptional events produced by cocaine conditioning. Finally, the largest number of cocaine-induced DEGs was found in the non-neuronal cells, including astrocytes and oligodendrocytes, most of which were not influenced by NPAS4, which is strictly neuronal. Astrocytes play an important role in shaping synaptic transmission associated with drug-cue reactivity[93], and the cocaine conditioning-induced changes in glial gene expression highlight the strong engagement of these non-neuronal populations in drug-related behavior.

Taken together, our findings reveal a role for NPAS4 in D2-MSNs in allowing cocaine, but not sucrose, to support prepotent drug-cue associations and relapse-like behavior. This is accomplished, at least in part, by blocking the cocaine experience-dependent strengthening of glutamatergic synaptic drive onto D2-MSNs, particularly from behaviorally required mPFC afferents, and by maintaining the appropriate activation balance of NAc D1- and D2-MSNs. NPAS4 is induced predominantly in NAc D1- and D2-MSNs following cocaine conditioning, and we showed that the ensemble of NPAS4-inducing neurons during cocaine conditioning is essential for the expression of cocaine CPP. NAc MSNs are functionally hyperpolarized and receive significant tonic inhibitory transmission from local interneurons under basal conditions[7]. Therefore, induction of NPAS4 in NAc D2-MSNs plays an essential homeostatic role to block synaptic plasticity that would perturb the D1:D2-MSN activation balance and reduce cocaine seeking. Understanding the mechanisms by which NPAS4 suppresses cocaine-induced neuroadaptations that oppose drug-cue/context associations might reveal new therapeutic avenues to block or reverse prepotent drug-cue associations that often trigger relapse in individuals recovering from SUD.

## Methods

### Animals
All experiments were conducted with male and female animals 8–16 weeks of age during the dark phase of their light/dark cycle. Targeted recombination in activated populations (TRAP) was achieved through design of a new "NPAS4-TRAP" mouse with NPAS4-dependent expression of cyclic recombinase that is fused with a triple mutant form of the human estrogen receptor (Cre-ERT2). Using these mice, NPAS4 is expressed along with Cre-ERT2, with a p2A cleavage site between that prevents changes in the function of endogenous NPAS4. When injected with a Cre-dependent AAV (see below), such as DIO-mCherry, cells expressing NPAS4 are labeled with mCherry following the injection of 4-hydroxytamoxifen (4OHT), which binds to ERT2 and allows the nuclear translocation and recombination of floxed genes by Cre.

Other experimental mice included C57BL/6J (wild-type, P60–80), heterozygous bacterial artificial chromosome (BAC) transgenic mice in which cyclic recombinase (Cre) was expressed under the control of the D1R promoter (Drd1a (D1)-Cre, line FK150) or the D2R promoter (Drd2 (D2)-Cre, line ER44), and BAC transgenic mice expressing tdTomato via the D1R promoter (D1-tdTomato) and eGFP via the D2R promoter (D2-eGFP). Mice were obtained from P. Kalivas (Medical University of South Carolina) and NINDS/GENSAT (http://www.gensat.org)[94,95]. Rats expressing Cre under the control of the D1R (D1-Cre) or D2R promoter (D2-Cre) were obtained from NIDA[55], who assisted us in the validation of Cre expression in D1- and D2-expressing MSNs, respectively, and then backcrossed to Charles River Long-Evans rats. Wild-type littermates were used for experimental controls. Both males and females were used for all experiments, and all animals were fed ad libitum, unless specified otherwise, and maintained at ~22 °C in a humidity-controlled environment on a reverse 12-h light/dark cycle (lights off at

8:00 a.m.). Animals were single-housed after surgery. All experiments were conducted in accordance with the US National Institutes of Health Guidelines for the Care and Use of Laboratory Animals and all procedures were approved by the Institutional Animal Care and Use Committee at the Medical University of South Carolina.

### Recombinant plasmids and shRNA expression viral vectors
For knockdown of endogenous *Npas4* mRNA expression in the NAc, previously validated *Npas4* shRNA (shNpas4) or non-targeting scrambled (shScram) shRNA control were cloned into the pAAV-shRNA vector as previously described[34,42]. The shNpas4 target sequence is GGTTGACCCTGATAATTTA, which is conserved in both the mouse and rat genomes. The adeno-associated virus serotype 2 (AAV2) vector consists of a CMV promoter driving eGFP with a SV40 polyadenylation signal, followed downstream by a U6 RNA polymerase III promoter and *Npas4* shRNA or scrambled (SC) shRNA oligonucleotides, then a polymerase III termination signal—all flanked by AAV2 inverted terminal repeats. For cell type-specific NPAS4 knockdown, *Npas4* shRNA or scrambled shRNA control was cloned into the pAAV-Sico vector[96] with slight modifications. The Cre-dependent *Npas4* shRNA viral vector has a constitutively-active promoter driving expression of mCherry, and a second, LoxP site flanked eGFP expression cassette that physically separates the U6 promoter from the *Npas4* shRNA coding sequence. In the absence of Cre recombinase, the virally-transduced cells express both mCherry and eGFP (but no shRNA). However, when this virus enters a cell that expresses Cre recombinase (e.g., D1-Cre animal), the eGFP expression cassette is excised and the cell expresses only mCherry and the *Npas4* shRNA (no eGFP). AAV2-*Npas4* shRNA, Cre-dependent *Npas4* shRNA, and scrambled control shRNAs were packaged by USC Vector Core (Columbia, SC). A second, sequence independent shNpas4 (shNpas4-2) was validated and prepared in the same way to confirm key behavioral findings; the sequence for this shRNA is GAACTGCTCTGTAAATCAT.

### Viral-mediated gene transfer
Stereotaxic surgery was performed under general anesthesia with isoflurane (induction 4–5% v/v, maintenance 1%–2% v/v). Coordinates to target the NAc (mainly core subregion) were +1.6 mm anterior, +1.5 mm lateral, and −4.4 mm ventral from bregma (relative to skull) at a 10 degree angle in mice, and +1.6 mm anterior, +2.8 mm lateral and −7.3 mm ventral (relative to skull) in rat. Coordinates to target the mouse mPFC prelimbic cortex were +1.85 mm anterior, +0.75 mm lateral, and −2.3 mm ventral from bregma at a 15 degree angle. AAV-*Npas4* shRNA, AAV-Cre-dependent *Npas4* shRNA, AAV-DIO-mCherry, AAV-DIO-GiDREADD-mCherry, AAV-Retro-Cre, and scrambled control shRNAs were delivered using Hamilton syringes or Nanoject III at a rate of 0.1 ul/min for a total 0.4–0.6 ul/hemisphere in mice and a total 0.6–0.8 ul/hemisphere in rats, with an additional 7–10 min before needles were retracted, as previously described[42,97]. Viral placements were confirmed by AAV-mediated fluorophore expression (i.e., GFP or mCherry) using confocal microscopy. See Supplemental Figs. S1C, S3G, S4I and S7B for representative viral placement and spread for each experiment.

### Immunohistochemistry
Mouse brains were drop-fixed overnight in 4% PFA in 1X PBS and transferred to a 30% sucrose solution in 1X PBS before slicing (40 or 50 mm) on a sliding microtome (Leica Microsystems). The slices were permeabilized and blocked in 3% BSA, 0.3% Triton X-100, 0.2% Tween-20, 3% normal goat serum in PBS, then incubated with primary antibodies: anti-NPAS4 (1:1000, kindly provided by Dr. Michael Greenberg's lab), anti-mCherry (1:500, chicken, LSBio), or anti-FOS (1:1000, rabbit, Synaptic Systems) in blocking buffer at room temperature for 2 h. Following a series of 1X PBS rinses, slices were incubated for 2 h at room temperature with secondary antibodies (donkey anti-rabbit 488,

donkey anti-rabbit 594, or donkey anti-chicken 594) while protected from light. Slices were counterstained with Hoechst, mounted with ProLong Gold, coverslipped on glass slides, and then analyzed by confocal microscopy (Zeiss LSM 880 or Leica Stellaris). Protein expression was measured using ImageJ software under experimenter-blinded conditions.

## Fluorescent in situ hybridization
Mice were live decapitated and brains were rapidly extracted and flash frozen in isopentane solution on dry ice. Brains were sliced at 16 μm in a cryostat (−18 °C), and sections were air-dried to the slides at −18 °C and stored at −80 °C. For the detection of D1 receptor (Drd1a), D2 receptor (Drd2), Cre recombinase, or *Npas4* mRNA, the RNAscope Fluorescent Multiplex kit (320293; Advanced Cell Diagnostics) was used with slight modification of manufacturer instructions and using commercially available probes [NPAS4 (C1, #423431), iCre (C2, #423321), Drd1a (C3, #406491), and Drd2 (C3, #406501)], as described previously[55]. Briefly, sections were fixed for 20 min in neutral buffered 10% formalin, followed by an ethanol dehydration series at 50%, 70%, 95%, and 100% for 5 min each. Before hybridization sections were treated with protease IV (322340, Advanced Cell Diagnostics) for 10 min. Sections were hybridized for 2 h at 40 °C, and probes were amplified using TSA fluorescent detection kits as directed by the manufacturer (Akoya Bioscience). The C1 probe was conjugated to Fluorescein, C2 to Cy3, and C3 to Cy5. Sections were then counterstained with DAPI, coverslipped using ProLong Gold mounting medium (Thermo Fisher), and allowed to dry. Slices were then imaged via confocal microscopy (Zeiss LSM 880).

## Mouse cocaine conditioned place preference
Mice were conditioned to cocaine using an unbiased paradigm, as described previously[42,98]. On day 1, mice were placed in the chamber and allowed to explore the conditioning apparatus, which consisted of two distinct environment chambers (white side or black side with different texture flooring). Mice that showed a pre-conditioning preference (more than 30% of the total time spent in either of the two chambers) were excluded from the study. On days 2 and 3, mice received cocaine (7.5 mg/kg; i.p.) in the AM and were confined to one chamber. In the PM sessions on days 2 and 3, 5–6 h after AM conditioning, mice received saline (0.9%; 1 ml/kg; i.p.) and were confined to the opposite chamber. On day 4, mice were placed again in one side of the CPP apparatus with free access to both chambers, and the time spent in each side was quantified. C57Bl/6J mice that were infused with AAV-Retro Cre in the NAc core and AAV-Cre-dependent Gi-DREADD mCherry in the PrL were subjected to a 6-day CPP paradigm, with cocaine pairings (7.5 mg/kg; i.p.) on days 2 and 4, and saline on days 3 and 5 on the opposite side of the place preference chamber. On the post-test (day 6), the mice were again placed in the CPP chamber with free access to both sides, and the time spent in each side was quantified. For all CPP tests, the cocaine and saline conditioning sessions lasted 30 min and pre- and post-test sessions lasted 20 min each. Data are expressed as the time spent on the cocaine-paired side minus the time spent on the saline-paired side (CPP score) during the post-test.

## Mouse locomotor sensitization
Locomotor activity was measured via photobeam array (San Diego Instruments) before and after each injection using the two-injection sensitization protocol[42,99]. Mice received saline injections for 3 days, then on the 4th day, mice received an injection of cocaine (20 mg/kg; i.p.). After 5 days, the mice received a 2nd cocaine injection (20 mg/kg; i.p.) in the locomotor apparatus. Data are shown as the sum of the total beam breaks in the first 30 min of each session.

## Rat cocaine or sucrose self-administration
Following Cre-dependent Npas4 shRNA or scrambled shRNA infusion into the NAc (as described above), rats were allowed to recover for

1 week prior to catheter implantation. Rat SA sessions (2 h) were performed during the light phase at the same time each day, during which the rats were placed in an operant conditioning chamber and connected to a drug line controlled by an external delivery pump. A "house" light inside the chamber signaled drug availability. All chambers contained an active (50 ul, 3 s infusion; 0.5 mg/kg cocaine) and an inactive lever. During an infusion, a cue light above the lever was illuminated and followed by a 20 s time-out period signaled by the house light going off. Rats self-administered cocaine on a fixed ratio (FR) schedule beginning at FR1, followed by FR3 and finally FR5. When FR requirements increased, intake was analyzed for stability (<20% variability within the last 3 sessions). Rats that completed 15–20 days of FR training and were stable on the FR5 schedule for at least 3 days continued to subsequent studies. Self-administration sessions (2 h) were performed at the same time each day during the dark phase as described previously[61]. Briefly, drug or sucrose availability was signaled by both the house light and a light above the active nose poke hole. Following a poke in the active hole, both availability lights were turned off and a cue light inside the nose poke hole was illuminated. Cocaine (12 ul, 2 s infusion; 0.5 mg/kg,) or a sucrose pellet (15 mg) were delivered immediately upon the active nosepoke, followed by a 10 s time-out period. Nose pokes in the inactive hole were without programmed consequences. Rats that met requirements for stable cocaine self-administration (last 3 days >85% discrimination between active and inactive nose poke holes and >10 infusions) entered into a 7-day abstinence phase. After this forced abstinence (withdrawal) period, the animals were placed back into the operant chambers and lever pressing in the absence of any drug administration, cues, and timeouts were measured on both levers. Extinction training continued for at least 6 days, followed by reinstatement sessions. Each reinstatement session consisted of a 2 h extinction session. Priming stimuli included presentation of the drug paired cue (light) or experimenter-administered cocaine (10 mg/kg, i.p.). Lever pressing was measured during the 2 h session. During cue-induced reinstatement, reward availability (house and active port light) was returned as an active lever pressing output, but drug or sucrose delivery was omitted.

## Dendritic spine morphometric analyses
D2-Cre mice received bilateral NAc injections of our Cre-dependent shNpas4 mixed 1:1 with a Cre-dependent spaghetti monster FLAG virus (AAV-CAG-Ruby2sm-FLAG-WPRE-SV40) to label D2-MSN dendritic spines using anti-FLAG IHC. Brains were collected after perfusion with 4% PFA in 1x PB, 24 h after the last cocaine conditioning session, fixed overnight in 4% PFA in 1x PB, then transferred to a 30% sucrose solution in 1x PB before slicing (100 μm) with a vibratome. D2-MSNs (expressing mCherry and 647-labeled FLAG) were sampled for dendritic spine analysis as described previously[100]. Briefly, dendrites past the secondary branch point and ≥100 um from the cell body were imaged using a Leica SP8 laser scanning confocal microscope equipped with HyD detectors for enhanced sensitivity. Dendritic spine segments were selected only if they satisfied the following criteria: (1) could clearly be traced back to a cell body of origin, (2) were not obfuscated by other dendrites, and (3) were co-labeled with somatic mCherry, but not eGFP, ensuring spines were sampled only in recombined, Cre+ neurons. Images were collected with a 63X oil immersion objective (1.4 N.A.) at 1024 × 512 frame size, 4.1X digital zoom, and a 0.1 μm Z-step size (0.04 × 0.04 × 0.1 μm voxel size). Pinhole was set at 0.8 airy units and held constant. The laser power and gain were empirically determined and then held relatively constant, adjusting only to avoid saturated voxels. Huygens Software (Scientific Volume Imaging, Hilversum NL) was used to deconvolve 3D Z-stacks. Deconvolved Z-stacks were then imported into Imaris (version 9.0.1) software (Bitplane, Zurich CH). The filament tool was then used to trace and assign the dendrite shaft. Dendritic spines were then semi-automatically traced using the autopath function, and an automatic threshold was used to

determine dendritic spine head diameter. Variables exported included the average spine head diameter (in μm) as well as the number of dendritic spines per μm of dendrite (spine density). Three to ten segments were sampled per animal, and the average spine head diameter and spine density were calculated for each segment. Data for each variable were then expressed as the number of spine segments/number of animals. All analyses were performed under experimenter-blinded conditions.

## Electrophysiology

All acute-slice electrophysiological experiments were performed in shScram and shNpas4 D1- or D2-Cre mice at 11–13 weeks old. Acute coronal slices (300-μm thickness) containing mPFC and NAc were prepared in a semi-frozen 300 mOsM dissection solution containing (in mM): 100.0 choline chloride, 2.5 KCl, 1.25 $Na_2H_2PO_4$, 25.0 $NaHCO_3$, 25.0 D-glucose, 3.1 Na-pyruvate, 9.0 Na-ascorbate, 7.0 $MgCl_2$, 0.5 $CaCl_2$ and 5.0 kynurenic acid (pH 7.4) and were continually equilibrated with 95% $O_2$ and 5% $CO_2$ prior to and during the slicing procedure. Slices were transferred to a 315 mOsM normal artificial cerebrospinal fluid (ACSF) solution containing (in mM): 127 NaCl, 2.5 KCl, 1.20 $Na_2H_2PO_4$, 24 $NaHCO_3$, 11 D-glucose, 1.2 $MgCl_2$, 2.40 $CaCl_2$, and 0.4 Na-ascorbate (pH 7.4) to recover at 37 °C for 30 min, and then transferred to room temperature ACSF for an additional 30 min prior to recording. NAc MSNs (depth 30–100 μm into the slice) were visualized using infrared differential interference contrast optics (DIC/infrared optics) and identified by their location, apical dendrites, and burst spiking patterns in response to depolarizing current injection. Cre-expressing cells (D1- or D2-MSNs) were identified by Cre-dependent viral expression of mCherry and the absence of GFP (which was removed in Cre-expressing cells to induce shRNA expression). Unless stated otherwise, all electrophysiological experiments were performed in whole-cell voltage clamp mode at −70 mV using borosilicate pipettes (4–6 MΩ) made on a NARISHIGE puller (NARISHIGE, PG10) from borosilicate tubing (Sutter Instruments) and filled with an internal solution containing (in mM): 140.0 CsMetSO4, 5.0 KCl, 1 $MgCl_2$, 0.1 $CaCl_2$, 0.2 EGTA, 11 HEPES, 2 NaATP, and 0.2 Na2GTP (pH 7.2; 290–295 mOsm).

All data (recordings) were acquired and analyzed using an AXO-PATCH 200B amplifier (Axon Instruments), a digitizer BNC2090 (National Instruments), AxoGraph v1.7.0, Clampfit v8.0 (pClamp, Molecular Devices), and MiniAnalysis Program v6.0.9 (Synaptosoft). Data were filtered at 2 kHz using an AXOPATCH 200B amplifier (Axon Instruments) and digitized at 20 kHz using AxoGraph v1.7.0.

## Optically-evoked postsynaptic currents

The evoked postsynaptic responses of NAc D1- or D2-MSNs were elicited by blue-light stimulation of excitatory afferents from the mPFC as previously described[72]. A 473 nm laser (Dragon Laser) coupled to a 50 μm core glass silica optical fiber (ThorLabs) was positioned directly above the slice orientated 30° ~350 μm from the recording electrode. At the site of recording discounting scattering a region of ~0.05 mm² was illuminated that after power attenuation due to adsorption and scattering in the tissue was calculated as ~100 mW/mm²[72,101]. Optically-evoked EPSCs were obtained every 10 s with pulses of 473 nm wavelength light (0–10 mW, 2 ms). AMPA-receptor-mediated excitatory postsynaptic currents (EPSCs) were recorded in the presence of picrotoxin (100 μM, Sigma-Aldrich) to block GABAARs. Outliers were assessed using the Grubbs test (alpha = 0.05) in GraphPad Prism software and were excluded from analysis.

## Spontaneous postsynaptic currents

Excitatory spontaneous postsynaptic currents (sEPSCs) were recorded from NAc MSNs in voltage clamp mode at −70 mV. MSNs were identified by their morphology parameters (medium-sized and round-shaped soma, spiny dendrites) and by bursting patterns of action potential firing in response to depolarizing current injection. For

recordings we chose specific Cre-expressing MSNs (D1- or D2) that were identified by Cre-dependent viral expression of mCherry and the absence of GFP (which was removed in Cre-expressing cells to induce shRNA expression). Data were recorded in a series of 10 traces (sweeps), for 10 s each. At the beginning of each sweep, a depolarizing step (4 mV for 100 ms) was generated to monitor the series (10–40 MΩ) and input resistance (>400 MΩ). To analyze the data, synaptic events were detected via custom parameters in the MiniAnalysis software (Synaptosoft, Decatur, GA) and subsequently confirmed by an observer. Data were measured until 700 events in a series were analyzed or until the maximal duration of the series.

## Single nuclei RNA-seq and bioinformatic analysis

Wild-type C57BL/6J mice received bilateral NAc infusion of AAV-*Npas4* shRNA-GFP or scrambled-GFP control and underwent conditioned place preference for cocaine or saline (a total of 4 groups with 4 animals per group). Mice were then rapidly decapitated at 12 weeks of age, and brains were extracted into 4 °C Hibernate A medium with Gluta-MAX, B27 supplement, and NxGen RNase inhibitor (0.2U/uL). NAc tissues were dissected rapidly, flash frozen, and stored at −80 °C until the day of dissociation. These specific steps were modified from ref. 49 by Hughes, B.W. The following day (when all samples were prepared), frozen tissue was slowly thawed on ice and chopped with a razor blade ~75–100 times in two orthogonal directions. Chopped NAc tissue was lysed with hypotonic lysis buffer (10 mM Tris-HCL, 10 mM NaCl, 3 mM $MgCl_2$, 0.1% (v/v) IGEPAL), supplemented with Hibernate-A medium, tissue triturated 15–20 times/sample using 3 glass capillaries with decreasing diameters, followed by 40-micron filtering. Nuclei were isolated by 500 × g centrifugation, washed with 1X PBS + 1% BSA and 0.2 U/ul RNase inhibitor. Nuclei were then incubated in eBioscience™ 7-Aminoactinomycin D (7AAD) viability staining solution (Thermo Fisher Cat #00-6993-50), which binds to DNA in permeabilized cells, allowing for the separation of clumps, debris, and whole cells from isolated nuclei during subsequent fluorescence-activated cell sorting (FACS). Samples were then counted and diluted to 1500 nuclei/μl before immediate processing using the 10x Genomics Single-Cell Protocol by the MUSC Translation Science Lab. Libraries were constructed using the Chromium Single-Cell 3′ Library Construction Kit (10x Genomics, v3.1) and sequenced using Vanderbilt's Next-Gen Sequencing Core (Illumina NovaSeq 6000).

## Sequencing analysis

Raw sequencing data were processed using Cell Ranger (v6.1.2) (PMID: 28091601). Cellranger mkfastq command was used to demultiplex the different samples and cellranger count command was used to generate gene–cell expression matrices. Ambient RNA contamination was inferred and removed using CellBender (v0.232) with standard parameters. Mouse genome mm10 was used for the alignment and gene-code vM25 was used for gene annotation and coordinates (PMID: 33270111).

Downstream analysis was performed in R with Seurat (v4.1.0) (PMID: 34062119) and customized R scripts. Data from different samples (S-scScram, C-scScram, S-scNPAS4, C- scNPAS4) were merged into a unique single-cell object. Nuclei with >250 genes, <10000 UMIs, and <5% mitochondrial transcripts were retained for downstream analysis. Genes located in the mitochondrial genome were excluded. Doublets were removed using scDblFinder (v1.8.0) (PMID: 35814628) to obtain an expression matrix with 21,516 genes and 52,743 cells. The SCTransform workflow was used for count normalization (VST transformation) and data integration (PMID: 31870423) using 30 principal components and resolution of 0.5 for Louvain clustering and UMAP. Cluster marker genes were identified using FindAllMarkers using the Wilcoxon Rank-Sum test with the standard parameters. After removing 4 clusters showing high expression of glutamatergic neuronal markers and subsequent re-clustering, we identified 20 distinct clusters. Cell

annotation was performed using two different approaches: (1) marker enrichment using Nucleus Accumbens independent data (PMID: 30257220) by Fisher's exact test and (2) a predictive machine learning model based on the Nucleus Accumbens atlas reference data (PMID: 34663959) using the R package scPred (v1.9.2) (PMID: 31829268). Cell annotation was then manually curated to reflect both analyses. Analysis for cell markers is provided in Supplementary Data 1 and 3.

### Differential expression analysis

To identify differentially expressed genes between conditions (S-shScram vs. C-shScram, C-shScram vs. C-shNPAS4) in each defined cell type, the R package LIBRA (v1.0.0) with the option MAST was used to perform zero-inflated regression analysis (PMID: 34584091). Genes were defined as significantly differentially expressed at Benjamini–Hochberg correction FDR < 0.05 and abs(log2(Fold Change)) > 0.2. We further confirmed these results using a Wilcoxon's rank-sum test. Analysis for DEGs is provided in Supplementary Data 2 and 4.

### Gene ontology analyses

Functional annotation of the identified DEGs was performed using ToppGene software (PMID: 19465376). A Benjamini–Hochberg FDR (FDR < 0.05) was used for multiple comparison adjustments. Enrichment was further confirmed using the R package clusterProfiler (v4.2.2) (PMID: 22455463).

### Statistics

Student's $t$ tests and two-way or three-way analyses of variance (ANOVAs) with or without repeated-measures (RM) were used, with ANOVAs followed by Sidak or Tukey post hoc tests when a significant interaction was revealed, to analyze mRNA expression, NPAS4 signal intensity, cocaine conditioned place preference, cocaine and sucrose self-administration, electrophysiology, and dendritic spine morphometric data. All statistical analyses were performed using GraphPad Prism, except for single nuclei RNA-sequencing analysis performed in R. Statistical outliers were detected using a Grubbs test and excluded from further analysis. All data are presented as mean ± SEM. Significance is shown as $^{\#}p < 0.1$, $^{*}p < 0.05$, $^{**}p < 0.01$, $^{***}p < 0.001$, $^{****}p < 0.0001$, and non-significant values are either not noted or shown as n.s. Detailed statistical results are provided in Supplementary Data 5.

### Reporting summary

Further information on research design is available in the Nature Portfolio Reporting Summary linked to this article.

## Data availability

All processed sequencing data, UMAP coordinates, and annotations have been made freely available to download at the BioCM portal: https://github.com/BioinformaticsMUSC/HughesEtAl_NPAS4Cocaine. Interactive sequencing data are available through ShinyCell at: https://bioinformatics-musc.shinyapps.io/Jessica_NAc_Cocaine_Npas4/. https://bioinformatics-musc.shinyapps.io/Hughes_NAc_Cocaine_NPAS4/. Raw and processed sequencing data to support the findings of this study have been deposited in GEO under accession number: GSE210850. Other source data are provided with this paper in the Source Data file. Source data are provided with this paper.

## Code availability

All code used to analyze the data and to generate the main figures of this paper can be found at[102]: https://github.com/BioinformaticsMUSC/HughesEtAl_NPAS4Cocaine/, https://doi.org/10.5281/zenodo.11265104.

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

## Acknowledgements

This work was supported by NIH grants F31 DA048557 (B.W.H.), T32 DA007288 (B.W.H., J.L.H., D.J.W.), NIH R01 DA032708 (C.W.C.), and the NIDA P50 DA046373 (C.W.C., M.T.). Bioinformatic analyses was supported by the Genomics and Bioinformatics Core of the NIGMS COBRE in Neurodevelopment and its Disorders (P20 GM148302). We also thank J. McGinty, P. Kalivas, J. Otis, E. Schmidt, T. Jhou, J. Woodward, C. Robinson, S. Duncan, and J. Day for advice on various aspects of the research designs and data analyses. We would also like to thank M.

Greenberg for sharing the anti-NPAS4 antibody. Some graphics and experimental design schema in Figs. 1, 2, 3, 5, 7 and S1, S3 were created with BioRender.com released under a Creative Commons Attribution-NonCommercial-NoDerivs (CC-BY-NC-ND) license. The opinions expressed in this article are the authors' own and do not reflect the views of the NIH/NIDA.

## Author contributions

B.W.H. contributed to the study design, data acquisition, data analysis, data interpretation, and manuscript preparation and revision. J.L.H. contributed to the data acquisition, data analysis, data interpretation, and manuscript preparation and revision. E.T., B.M.S., K.K.S., R.M.A., and D.J.W. contributed to the data acquisition and analysis. R.D.P., M.D.S., and S.B. contributed to study design, data acquisition, data analysis, data interpretation, and manuscript revision. M.T. contributed to AAV vector design and production, data interpretation, and manuscript preparation and revision. C.W.C. contributed to study design, data analysis, data interpretation, and manuscript preparation and revision. All authors critically reviewed the content and approved the final version for publication.

## Competing interests

The authors declare no competing interests.
