## [Peer Review File · Nature Communications]

NPAS4 supports cocaine-conditioned cues in rodents by controlling the cell type-specific activation balance in the nucleus accumbensEditorial Note: Parts of this peer review file have been redacted as indicated to remove confidential information.

REVIEWER COMMENTS

Reviewer #1 (Remarks to the Author):

In this manuscript the authors examine the cell-type specific effects of Npas4 induction by cocaine in cells of the mouse nucleus accumbens. To determine the function of the cells that induce Npas4, the authors use an Npas4-TRAP system to tag cells that activate Npas4 during the association of phase of cocaine CPP and then show that silencing of this ensemble suppresses the CPP preference score. They use single cell sequencing to show a role for Npas4 in D2R expressing neurons, and then validate that Npas4 KD in D2R neurons (but not D1R neurons) also disrupts cocaine-induced behaviors. Interestingly they find that KD of Npas4 in D2R neurons leads to enhanced cocaine-induced Fos induction in these neurons suggesting that they fire more with cocaine than under control conditions. Further studies show that Npas4 KD in D2R neurons results in changes in gene expression, increased spine density, and enhanced glutamatergic synapse connectivity inferred from the response to optogenetic stimulation of inputs. These data overall suggest that Npas4 induction by cocaine functions in D2R MSNs to promote cocaine seeking, and that getting rid of Npas4 would be anti-addictive. This is consistent with a prior study from the same lab showing that conditional viral deletion of Npas4 in the NAc impairs cocaine CPP.

A strength of this study is localizing at least some of the actions of Npas4 to D2R neurons in the NAc and showing that a plausible mechanism for Npas4 in these cells is driving excitatory input to these neurons. However, we do have some concerns we would like the authors to address.

- The evidence that there is a difference between knocking down Npas4 in D2R neurons of the NAc versus D1R neurons is largely compelling as far as what is presented. However, since the KD is the crux of the study, the authors need to replicate at least their major finding with a second sequence independent shRNA to verify that these results are not due to off-target effects.
- The one piece of behavioral data that is not totally convincing is the cue-induced reinstatement in the D1- and D2-Cre rats. The scrambled lever presses are what are really different between the two strains, which raises the question of whether the D2 "specific" effect is really just a secondary consequence of differences between the strains rather than something about cell type in this behavior.

With respect to the FANS data:

- The authors never specify the method used for the fluorescence-based nuclear sorting. Is this just via incubation with DAPI to get all nuclei? Or are the authors only sorting and sequencing the cells that got infected with the virus? If so, which fluorophore is being used, and how does that efficiently isolate nuclei? These methods need to be made much more clear.
- Other than performing GO analysis, the authors don't really use the sequencing data from the knockdown conditions. Genes are significantly different in KD in both D1R and D2R cells. What do the authors make of this? Why would Npas4 KD have any effect in D1R cells if it is not expressed there? What genes in the D2R cells do the authors think may contribute to the connectivity phenotype?
- Is Figure 2b all of the nuclei from all of the conditions clustered together? Does this mean that cell type is a much stronger driver of clustering than any change in gene expression due to KD or cocaine? What has been done in the method and what is shown in the figure needs to be much more clearly explained.
- It does not make sense to split the snRNAseq data between Figure 2 and 5. The data fit much more with Figure 5, as they do not really serve as a control for the TRAP to show where Npas4 is induced (as mentioned above). It makes more sense to present the data together in the position of Figure 5.

About the data showing Npas4 induction in D2R neurons:

- In a previous study using an Npas4 Ab that is used, though very sparingly, in this study, the same laboratory showed that rapidly following cocaine only 15% of Npas4 protein induction was in DARPP32 neurons, whereas 85% appeared to be in other types of cells that were suggested to be interneurons. Here the authors use a very different type of tool to mark Npas4 induction that reflects transcription of the NPas4 gene but could be independent of its translation. Did the authors use the Npas4 antibody to verify that they can see Npas4 induction in the Npas4 trapped ensemble if they give cocaine again after the CPP labeling? This appears to be what is attempted in Extended data 1h, but the method and the result was not easy to follow in the text. Regardless of the result the authors need to discuss the difference in their cell type focus in this paper and that one and how they reconcile the two studies. Are interneurons also labeled by the TRAP method here?
- Npas4 RNA is very rapidly induced and degraded following cocaine or other stimuli, thus what do the authors think is the meaning of identifying Npas4 RNA in a very small fraction of D2Rs by scRNAseq 24 hours after the final exposure to the drug?

About the TRAP controls:

- What exactly is the Saline TRAP? Is this an ensemble for the novel chamber? Is the number of trapped cells similar to the number after cocaine?
- It would have been useful if the authors had sequenced the TRAPPED neurons.

Minor concerns:

- The point of validating the D2R/GRM8 cells was not clear.
- Figure 3 is worded wrong in that they mix up mice and rat references both in the caption and the text
- Figure 3 F, G, H, and I all need better legends to explain colors
- Figure 4 Figure e is not labelled well, and ventral is spelled wrong

Reviewer #2 (Remarks to the Author):

Hughes et al investigated that impact of Npas4 in the D1 and D2-MSNs on drug-cue association and relapse-like behaviors of mice. The authors developed Nap4-TRAP mice and characterized at single cell level. Subsequently, showed the cell-type specific activation of Npas4 into D2-MSN plays a crucial role to promote drug-cue association and relapse. The most critical issue is that impact of present study is weak because it remains unclear the molecular mechanism by which Npas4 regulates drug-cue association although each experiment is well conducted and documented.

The authors claimed an importance of Npas4 in D2-MSN. Neuronal activity of D2-MSN of ShNpas4 during CPP or reinstatement should be monitored to show the functional significance.

In Fig. 1, CPP score of saline TRAP mice with or without CNO is high score. To show that sensitivity of CPP induced by cocaine was detectable, saline TRAP and cocaine TRAP should be compared directly in a same figure. Please combine Fig. 1c and e as new figure 1c.

The authors found that 75% of Npas4-positive neurons expressed either D1R (49%) or D2R (27%) during cocaine conditioning (Fig. 1F). This suggests that the majority of cells expressing Npas4 during cocaine conditioning are D1R-MSN or D2R-MSN. However, the author's group (Taniguchi et al.) have previously reported that only 15% of Npas4-positive neurons colocalized with MSN markers during cocaine conditioning (Neuron. 2017 Sep 27;96(1):130-144.e6. doi: 10.1016/j.neuron.2017.09.015.). I am wondering if the 75% of Npas4-positive neurons expressing D1R or D2R are MSNs other type of cells. The authors need to clarify this discrepancy in the text of the revised manuscript.

Percentage of cocaine-TRAPed Npas4-positive neuron was shown in figure 1f. Half of cocaine-TRAPed Npas4-positive neuron was assigned in to D1R. However, knock down of Npas4 in D1-MSN had no effect on CPP score and cue-induced reinstatement in Fig. 3c and f. Funahshi et al. have reported that deletion of Npas4 in D1-MSNs impairs cocaine-induced place preference (Cell Rep. 2019 Dec 3;29(10):3235-3252.e9. doi: 10.1016/j.celrep.2019.10.116.). Please discuss this discrepancy in the main text of revised manuscript.

Single nuclei RNA sequencing was carried out in shNpas4 mice. I am wondering why the authors selected shNpas4 although conditional Npas4 KO mice was available. Knock down efficacy of shNpas4 should also be provided.

Cue-induced reinstatement was affected by motivational property in Fig. 3. Please measure breaking points of progressive schedule task.

The authors found that knockdown of Npas4 in D2R-MSNs significantly reduced the number of presumed FOS-positive neurons in D1-MSNs (Fig. 4D, right). The authors need to explain why knockdown of Npas4 in D2R-MSNs reduces FOS-positive neurons in D1R-MSNs. Does the increase of neural activity in D2R-MSN caused by ShNpas4 affect neural activity in the D1R-MSN? The authors need to prove this experimentally.

In Fig. 4, when Npas4 is knocked down using D1R-Cre mice, it is incorrect to describe the cells that were not knocked down as D2R-MSN. Similarly, when Npas4 is knocked down using D2R-Cre mice, it is incorrect to describe the cells that were not knocked down as D1R-MSN. The authors should state this correctly, as other subtypes of cells may be included.

The authors measured the number of Fos-positive cells in dorsolateral VP in Fig. 4. To show direct evidence the accumbal D2-MSNs to dorsolateral VP pathway was regulated by Npas4 in D2-MSN. Connectivity between accumbal D2-MSNs and Fos-positive neuron in dorsolateral VP of shScram_D2 and ShNpas4_D2 should be monitored by tracing technique.

The authors found that Npas4 knockdown in mice conditioned to cocaine, but not saline, produced a significant increase in D2-MSN dendritic spine density in Fig. 6B and C. Lissek et al. have recently reported that Npas4 knockdown decreased spine density under basal conditions and prevented the cocaine-evoked increase in dendritic spine density in D1R-MSN. Whereas there were no significant differences in spine density in any treatment group in D2R-MSNs, indicating that Npas4-mediated regulation of spine density is restricted to D1-MSNs (EMBO Reports. 2021 Dec 6;22(12):e51882. doi: 10.15252/embr.202051882.). The authors should resolve this critical discrepancy in the main text of revised manuscript.

Although almost all experiments in this study have been carried out with a single knockdown vector, the use of multiple siRNAs against the same target, experiments to rescue RNAi effects by expressing siRNA-resistant genes or the use of knockout mice are required to increase the reliability of the data.

Minor point

It is unclear the injected position of each animal. In explaining AAV experiments the authors should refer more accurately to the site of injection. The authors should also provide a composite drawing of the injection sites based on fluorescence in their slices in the supplement in order to validate a consistent injection site.

In Fig 3A, which animal did the authors use, mice or rats? The authors describe mice in the main text, but rats in Figure legend.

The authors described Penk and Tac1 in the main text. There was no figure panel regarding Penk and Tac1 in Fig. 5e.

The authors need to describe the measurement time of the CPP post-test in Materials and Methods section.

For the graph of CPP scores, not only the post-test values should be shown, but also the pre-test values.

As with other Figs, the graphs in Figs 4c, 4d, 4h, 4i and 6c should be presented as bar and scatter plots.

Reviewer #3 (Remarks to the Author):

The study described in this manuscript is a tour de force. It identifies a host of new properties of the transcription factor NPAS4 and how these properties play a role in cocaine-induced changes in brain and behavior. The study makes use of a wide variety of powerful approaches to address key issues in NPAS4 activation (e.g. when, where and the downstream consequences for neuronal activity and behavior), including the introduction and validation of an NPAS4-TRAp2a mouse line that should be very useful for future studies in the field. The results in the manuscript bring up lots of new and interesting questions that presage some exciting work ahead. I have several concerns about the manuscript in its present form, each of which can be readily addressed.

1. Although it is cited, there is scant discussion (unless I somehow missed it?) of the Lissek et al. 2021 study examining NPAS4's role in shaping neuronal excitability in accumbens MSNs and cocaine-induced behavior as it relates to the current study. This is warranted, especially as the results appear to be slightly at odds with those in the current study?
2. The study uses both female and male mice. This is as it should be, of course. At the same time there seems to be no statement of whether the data were examined with respect to sex and whether any differences were noted. The study may not have been powered to detect subtle differences but given that the data in this manuscript will likely set the stage for a wide array of future studies on the topic and that the literature is now replete with data indicating sex-differences in neural factors that contribute to reward processing and behavior, this should be addressed in some way.
3. (Minor) The title should include "cocaine" as this was the drug used to test the cue- and relapse-like-related behaviors and different drugs can engage different neural elements.
4. (Minor) There are quite a few typos throughout the manuscript (e.g. repeated words, incorrect tenses, etc.) that can get cleaned up.

Response to Reviewers

Hughes, Huebschman et al.

02/07/2024

We thank the reviewers for their insightful comments. We are submitting a revised version of our manuscript on NPAS4's role in supporting cocaine-cue associations and relapse-like behavior for consideration at Nature Communications and hope you will agree that we have reasonably addressed all advice and concerns. Importantly, we now include an additional single nuclei RNA sequencing (snRNA-seq) experiment which provides further data on the identity of the cell types included in the *Npas4*+ ensemble, and we have replicated our main behavioral finding that NPAS4 functions in NAc D2-expressing cells using an additional, validated *Npas4* shRNA, the latter of which has taken nearly a year to generate, validate, and perform behavior in D2-Cre mice. We have also disaggregated data by sex, where appropriate, and include a more complete consideration of sex as a biological variable. Copied below are reviewer comments and our responses (in blue).

Reviewer #1 (Remarks to the Author):

In this manuscript the authors examine the cell-type specific effects of *Npas4* induction by cocaine in cells of the mouse nucleus accumbens. To determine the function of the cells that induce *Npas4*, the authors use an *Npas4*-TRAP system to tag cells that activate *Npas4* during the association of phase of cocaine CPP and then show that silencing of this ensemble suppresses the CPP preference score. They use single cell sequencing to show a role for *Npas4* in D2R expressing neurons, and then validate that *Npas4* KD in D2R neurons (but not D1R neurons) also disrupts cocaine-induced behaviors. Interestingly they find that KD of *Npas4* in D2R neurons leads to enhanced cocaine-induced Fos induction in these neurons suggesting that they fire more with cocaine than under control conditions. Further studies show that *Npas4* KD in D2R neurons results in changes in gene expression, increased spine density, and enhanced glutamatergic synapse connectivity inferred from the response to optogenetic stimulation of inputs. These data overall suggest that *Npas4* induction by cocaine functions in D2R MSNs to promote cocaine seeking, and that getting rid of *Npas4* would be anti-addictive. This is consistent with a prior study from the same lab showing that conditional viral deletion of *Npas4* in the NAc impairs cocaine CPP.

A strength of this study is localizing at least some of the actions of *Npas4* to D2R neurons in the NAc and showing that a plausible mechanism for *Npas4* in these cells is driving excitatory input to these neurons. However, we do have some concerns we would like the authors to address.

- The evidence that there is a difference between knocking down *Npas4* in D2R neurons of the NAc versus D1R neurons is largely compelling as far as what is presented. However, since the KD is the crux of the study, the authors need to replicate at least their major finding with a second sequence independent shRNA to verify that these results are not due to off-target effects.

We thank the reviewer for agreeing that the cell type-specific knockdown data are compelling, but that confidence is further increased with a second approach. In our prior publication (Taniguchi et al, 2017, Neuron), we had used both AAV2-shNPAS4 and AAV2-Cre injected into the NAc of *Npas4*-floxed mutant mice, which gave similar results, thus providing confidence that our shRNA effects are specific. However, since those original experiments were not cell type-specific, we have designed and validated a Cre-dependent version of the original shRNA, and a second independent shRNA targeting *Npas4* mRNA for degradation. We now show that use of this second AAV2-shNPAS4 in NAc of D2-Cre mice replicates the decreases in cocaine CPP produced by the original AAV2-SICO-shNPAS4 (see new Figures S3L-O). The design, creation, viral prep and validation, and adequately powered cocaine CPP study have taken nearly one year to complete, and we hope the reviewers will agree that the results provide strong confidence in the D2-specific effects of AAV2-SICO-shNpas4-2.

- The one piece of behavioral data that is not totally convincing is the cue-induced reinstatement in the D1- and D2-Cre rats. The scrambled lever presses are what are really different between the two strains, which raises the question of whether the D2 “specific” effect is really just a secondary consequence of differences between the strains rather than something about cell type in this behavior.

The reviewer makes a good point, and there are documented instances of strain-specific differences in behavior between D1-Cre and D2-Cre rats. Of note, however, is that despite similar differences in the scrambled control groups during primed reinstatement (Figure S4D,H), we observe no effect of shNPAS4 in either group in this case. Further, shNpas4 appears to *increase* cue-induced reinstatement of sucrose seeking in D2-Cre rats (Figure S4O). This, alongside our findings of D2- specific effects in a mouse model, increases our confidence that the observed effects are due to knockdown of *Npas4* in this cell type, rather than a secondary consequence of strain differences.

With respect to the FANS data:

- The authors never specify the method used for the fluorescence-based nuclear sorting. Is this just via incubation with DAPI to get all nuclei? Or are the authors only sorting and sequencing the cells that got infected with the virus? If so, which flurophore is being used, and how does that efficiently isolate nuclei? These methods need to be made much more clear.

We used 7AAD to isolate nuclei via FANs (7AAD+ nuclei no longer have a cellular membrane and are, therefore, considered “dead”); we are not sorting for viral expression. We have edited the text in the methods section and throughout the manuscript where appropriate for clarity.

- Other than performing GO analysis, the authors don’t really use the sequencing data from the knockdown conditions. Genes are significantly different in KD in both D1R and D2R cells. What do the authors make of this? Why would *Npas4* KD have any effect in

D1R cells if it is not expressed there? What genes in the D2R cells do the authors think may contribute to the connectivity phenotype?

To clarify, this shNpas4 snRNA-sequencing experiment (now Figures 5 and 6) was conducted in wild-type mice with constitutive expression of shNpas4 or shScram in the NAc, so knockdown was not cell type-specific. Given the cell type-specific behavioral effects we observed, we chose to conduct targeted analysis within cell type clusters for this sequencing experiment. We now confirm in two independent snRNA-sequencing experiments (Figures 2 and 5-6), and via cocaine experience-induced NPAS4 IHC in D1-tdTomato x D2-eGFP mice, that Npas4 mRNA and protein are both expressed in D1R and D2R cells, suggesting that DEGs in each of these cell types could be a direct effect of Npas4 knockdown. It is also possible, as the reviewer suggests, that some of these DEGs are secondary changes resulting from changes in cell connectivity and/or activity levels driven by the direct effects of Npas4 knockdown, though we feel this question is beyond the scope of the current work. We have clarified the details of the constitutive shNpas4 snRNA-sequencing experiment in the text.

- Is Figure 2b all of the nuclei from all of the conditions clustered together? Does this mean that cell type is a much stronger driver of clustering than any change in gene expression due to KD or cocaine? What has been done in the method and what is shown in the figure needs to be much more clearly explained.

Yes, the original Figure 2B (now Figure 5B) includes all of the conditions with clusters generated specifically using canonical cell type-specific markers. We see that *Npas4* expression and/or cocaine exposure has minimal effect on the expression of these canonical markers and is therefore not a large driver of clustering, in this case. However, this doesn't necessarily mean that differential gene expression due to knockdown or cocaine would not be a strong driver of clustering, if other parameters were used to guide clustering. We chose to cluster based on cell types first, to confirm we had a population representative of the NAc and to facilitate analysis of differential gene expression within specific cell type populations. This same clustering approach was taken for our new snRNA-Seq experiment in the new Figure 2C. We also show UMAPs for each individual group/condition, showing no strong shift in cluster following either cocaine conditioning or Npas4 knockdown. We have clarified our approach in the methods, figures, and other relevant sections of the text.

- It does not make sense to split the snRNAseq data between Figure 2 and 5. The data fit much more with Figure 5, as they do not really serve as a control for the TRAP to show where Npas4 is induced (as mentioned above). It makes more sense to present the data together in the position of Figure 5.

We thank the reviewer for this helpful advice. We now include an additional snRNA-seq experiment that aims to better define the Npas4+ ensemble and have made this Figure 2. The original data the reviewer references in this comment (above) are now shown sequentially in Figures 5 and 6.

About the data showing Npas4 induction in D2R neurons:

- In a previous study using an Npas4 Ab that is used, though very sparingly, in this study, the same laboratory showed that rapidly following cocaine only 15% of Npas4 protein induction was in DARPP32 neurons, whereas 85% appeared to be in other types of cells that were suggested to be interneurons. Here the authors use a very different type of tool to mark Npas4 induction that reflects transcription of the NPas4 gene but could be independent of its translation. Did the authors use the Npas4 antibody to verify that they can see Npas4 induction in the Npas4 trapped ensemble if they give cocaine again after the CPP labeling? This appears to be what is attempted in Extended data 1h, but the method and the result was not easy to follow in the text. Regardless of the result the authors need to discuss the difference in their cell type focus in this paper and that one and how they reconcile the two studies. Are interneurons also labeled by the TRAP method here?

The reviewer raises a few important points here, which we address below:

- First, there are differences between Npas4 RNA and NPAS4 protein expression, as our sequencing experiments suggest low basal levels of RNA expression in a number of cells (~8.5%), but we and others have found few to no NPAS4+ cells when looking at protein expression in naïve animals. The new snRNAseq experiment included in the revised manuscript (Fig. 2) supports our original RNAscope finding in the TRAP mouse that the majority (~75%) of Npas4 mRNA positive cells during cocaine conditioning are MSNs, increasing our confidence in this conclusion.
- Second, as the reviewer mentioned, our lab previously reported ~15% of cells positive for NPAS4 protein expression were also positive for the MSN marker DARPP32 and that ~25% were also positive for interneuron markers (GAD67, PV, SST). We suspect that the underestimation of the MSN population in the prior publication is due to sub-par immunodetection with the DARPP32 antibody. To complement the anti-NPAS4 rabbit polyclonal, we had to use a mouse monoclonal anti-DARPP32 antibody that likely underestimated the number of NPAS4+ MSNs. In the revised manuscript, we now include new IHC analysis of anti-NPAS4 in Drd1a-tdTomato/Drd2-eGFP mice, which revealed that ~50% of the NPAS4+ cells are D1R or D2R+. This distribution is still lower than the mRNA analysis (snRNA-seq or RNAscope), which could be due to subthreshold detection of tdTomato/EGFP signal or due to differences in NPAS4 protein synthesis; we see very similar results following sucrose SA in mice (data not in publication). Our revised manuscript now includes the new anti-NPAS4 IHC in Drd1-tdTomato;Drd2-eGFP mice (Figure S2B), as well as new snRNA-seq data that assesses NPAS4 in all NAc cell populations. Of note, a significant fraction of Npas4+ neurons appear to be Grm8+, non-canonical MSNs that express very low or undetectable levels of Drd1 or Drd2 mRNAs. The distribution of Npas4+ neurons are now clearly shown in the revised manuscript.

- Third, the reviewer is correct about the experiment in Extended Data 1H-I, where we find that approximately 40% of NPAS4-TRAP cells are positive for NPAS4 protein expression after an additional exposure to cocaine. It is difficult to say from this experiment whether our finding is due to a refinement of the ensemble with repeated exposure or an inherent difference in the number of cells that induce RNA versus protein expression, but this reactivation pattern is similar to what has been observed for FOS in Fos-TRAP cells (see Denardo et al, 2019) and ARC in Arc-TRAP cells (personal communication).
- Finally, we do detect NPAS4 mRNA expression in several interneuron types, including Pvalb+ interneurons. The NPAS4+ cell types are all now reported in the new Fig. 2.

In the revised manuscript, we now include additional discussion on the discrepancies between our current *Npas4* ensemble characteristics and the IHC findings in our 2017 paper (*Neuron*). We have also clarified the reactivation experiment from Extended Data 1H-I.

- *Npas4* RNA is very rapidly induced and degraded following cocaine or other stimuli, thus what do the authors think is the meaning of identifying *Npas4* RNA in a very small fraction of D2Rs by scRNAseq 24 hours after the final exposure to the drug?

This is an interesting point, perhaps furthered by our new snRNA-seq experiment (see Figure 2). We observe a similar percentage of *Npas4*+ cells in NAc tissue collected 15 minutes (a time-point shown to represent peak *Npas4* mRNA expression) after cocaine exposure compared to unstimulated, cocaine-naïve mice. We think that *Npas4* mRNA must be expressed basally at low levels in these cells, and then following cocaine conditioning, *Npas4* mRNA is rapidly induced in a small subset of NAc neurons. To support this idea, the same sequencing data indicates *Npas4* is significantly upregulated in D1- and D2-MSNs (as well as a few other neuron-specific cell types), and that there is a higher percentage of “highly” expressing *Npas4*+ cells overall, in cocaine-conditioned animals compared to the home-cage controls. With this in mind, we think it is likely that the *Npas4* expression observed at the 24-hour time point is reflective of a return to baseline levels. We also see similar numbers of NPAS4 protein-expressing cells when tagging ensembles during saline vs. cocaine CPP, reiterating this point. The revised manuscript now includes discussions of basal levels of *Npas4* expression vs. induced, and we have also clarified that the 24-hour time point in our original experiment was selected to identify transcriptome changes (and presumed downstream effects of NPAS4) that are present at the time of expression of CPP behavior, the timing of analysis of spine density, and PrL→NAc D2-MSN oEPSC effects (i.e. Late Response Genes).

About the TRAP controls:

- What exactly is the Saline TRAP? Is this an ensemble for the novel chamber? Is the number of trapped cells similar to the number after cocaine?

Yes, the number of trapped cells during saline conditioning is similar to the number after cocaine conditioning (data included in Extended Data Fig. 1G). We agree with the reviewer in that it seems likely that the saline-conditioned ensemble encodes information about the stimulus (i.p. saline injection) and the conditioning chamber. We suspect that is why inhibiting the saline ensemble doesn't alter cocaine CPP, since it likely encodes the saline-context experience. Future studies will be important to further understand whether the saline-context ensemble encodes specific information about the saline control context. We have added clarifying points in the text regarding the saline TRAP experiment.

- It would have been useful if the authors had sequenced the TRAPPED neurons.

We agree with the reviewer and have attempted this multiple times without success. To help define the NPAS4 ensemble, we instead treated mice with or without cocaine conditioning, and examined the NPAS4+ populations at 15 mins post conditioning in a new snRNA-seq experiment that is included in the revised manuscript (see Figure 2). Future studies will seek to isolate and sequence the <1% TRAP ensemble, but we hope the reviewer will agree that this is beyond a reasonable scope for this data-rich manuscript.

Minor concerns:

- The point of validating the D2R/GRM8 cells was not clear.

We wanted to confirm our snRNA-seq data that indicated *Npas4* was abundant in the *Grm8+* cells, as this is a relatively novel and non-canonical MSN subtype. We have clarified this point in the text.

- Figure 3 is worded wrong in that they mix up mice and rat references both in the caption and the text

We have corrected this and added additional clarification on the use of mice (for the CPP experiments) and rats (for the SA experiments).

- Figure 3 F, G, H, and I all need better legends to explain colors

We have changed the legends for clarity as suggested.

- Figure 4 Figure e is not labelled well, and ventral is spelled wrong

We have corrected this.

Reviewer #2 (Remarks to the Author):

Hughes et al investigated that impact of Npas4 in the D1 and D2-MSNs on drug-cue association and relapse-like behaviors of mice. The authors developed Nap4-TRAP mice and characterized at single cell level. Subsequently, showed the cell-type specific activation of Npas4 into D2-MSN plays a crucial role to promote drug-cue association and relapse. The most critical issue is that impact of present study is weak because it remains unclear the molecular mechanism by which Npas4 regulates drug-cue association although each experiment is well conducted and documented.

The authors claimed an importance of Npas4 in D2-MSN. Neuronal activity of D2-MSN of ShNpas4 during CPP or reinstatement should be monitored to show the functional significance.

We agree and attempted to begin answering this question by looking at FOS expression (Figure 4) after a cocaine conditioning session during which time an animal is forming drug-context associations presumably important during CPP. We are working on establishing methods in our laboratory for *in-vivo* calcium imaging in behaving animals which will aid in our future investigation of this question, but we feel that head-fixed, *in vivo* calcium imaging is beyond a reasonable scope for this current study.

In Fig. 1, CPP score of saline TRAP mice with or without CNO is high score. To show that sensitivity of CPP induced by cocaine was detectable, saline TRAP and cocaine TRAP should be compared directly in a same figure. Please combine Fig. 1c and e as new figure 1c.

To clarify, both figure 1c and 1e are showing the post-test scores for the cocaine CPP paradigm, not saline CPP. We would expect high scores in both cases as we are looking at time spent in the cocaine paired chamber in both, and directly comparing the two as suggested would not provide any information on the sensitivity of CPP induced by cocaine vs saline. Rather, the difference between the two experiments in 1c and 1e is related to when 4OHT was administered during conditioning (during cocaine sessions vs during saline sessions). We realize this may have been confusing and have clarified the text and figure legend to make this distinction clearer. We thank the reviewer for bringing this to our attention and hope our clarification alleviates any concerns regarding the CPP scores in the saline TRAP experiment, in addition to the pre-test scores shown in the Supplement.

The authors found that 75% of Npas4-positive neurons expressed either D1R (49%) or D2R (27%) during cocaine conditioning (Fig. 1F). This suggests that the majority of cells expressing Npas4 during cocaine conditioning are D1R-MSN or D2R-MSN. However, the author's group (Taniguchi et al.) have previously reported that only 15% of Npas4-positive neurons colocalized with MSN markers during cocaine conditioning (Neuron. 2017 Sep 27;96(1):130-144.e6. doi: 10.1016/j.neuron.2017.09.015.). I am wondering if the 75% of Npas4-positive neurons expressing D1R or D2R are MSNs other type of cells. The authors need to clarify this discrepancy in the text of the revised manuscript.

First, we do think there is a discrepancy between Npas4 RNA and NPAS4 protein expression, as our sequencing experiments suggest low basal levels of RNA expression in a number of cells (~8.5%) but we and others have found few to no NPAS4+ cells when looking at protein expression in naïve animals. The additional snRNAseq experiment included in the revised manuscript (Figure 2B-G) supports our original RNAscope finding in the TRAP mouse that the majority (~75%) of Npas4 RNA positive cells during cocaine conditioning are MSNs (now Figure 2A), increasing our confidence in this conclusion.

Our lab previously reported ~15% of cells positive for NPAS4 protein expression were also positive for the MSN marker DARPP32, and that ~25% were also positive for interneuron markers (GAD67, PV, SST). However, we also reported that ~80% of these NPAS4+ cells were neurons (NEUN positive), leaving the cell type of a significant population of NPAS4+ neurons undefined. We think it possible that this undefined population consisted of MSNs which were poorly labelled by the DARPP32 antibody used in that study and have more recently begun using other approaches and/or markers for quantification of MSNs. Our revised manuscript now also includes NPAS4 IHC in D1-tdTomato x D2-eGFP mice (Figure S2B), where we report ~50% of NPAS4+ cells are D1R or D2R cells, where NPAS4 protein expression was assessed at its peak following cocaine conditioning.

We have added additional discussion of these findings in the text.

Percentage of cocaine-TRAPed Npas4-positive neuron was shown in figure 1f. Half of cocaine-TRAPed Npas4-positive neuron was assigned in to D1R. However, knock down of Npas4 in D1-MSN had no effect on CPP score and cue-induced reinstatement in Fig. 3c and f. Funahashi et al. have reported that deletion of Npas4 in D1-MSNs impairs cocaine-induced place preference (Cell Rep. 2019 Dec 3;29(10):3235-3252.e9. doi: 10.1016/j.celrep.2019.10.116.). Please discuss this discrepancy in the main text of revised manuscript.

Funahashi *et al* utilized a dominant-negative overexpression approach to interfere with NPAS4 function, whereas we utilize an shRNA knockdown approach. Further, our CPP studies were conducted with a 7.5 mg/kg dose, whereas Funahashi et al utilize a 10 mg/kg dose. We suspect that these differences might contribute to the varying results, but it's also possible that the Npas4 dominant-negative (isolated DNA binding domain) might produce non-specific effects that are distinct from reducing levels. We also note that the group did not validate that their FLEX-NPAS4 Dom Neg was restricted in expression to D1R+ cells in the *Drd1*-Cre mice. We now discuss the discrepancy between these two studies in the revised discussion, and future studies examining the impact of the NPAS4 DBD vs. our two shRNAs will be needed to fully resolve this discrepancy.

Single nuclei RNA sequencing was carried out in shNpas4 mice. I am wondering why the authors selected shNpas4 although conditional Npas4 KO mice was available. Knock down efficacy of shNpas4 should also be provided.

We utilized an shNpas4 approach in our initial behavioral studies in order to induce a region (NAc)- and cell type-specific knockdown effect. In the snRNA-seq experiment, we opted to use a similar approach to maintain consistency in knockdown (compared to the majority of results in the paper; i.e. ephys, spine density analysis), particularly as we were investigating transcriptomic differences at a timepoint relevant to our behavioral findings. Knockdown efficiency is provided in Figure S3.

Cue-induced reinstatement was affected by motivational property in Fig. 3. Please measure breaking points of progressive schedule task.

We think it important to note that while we observe an effect on cue-induced reinstatement, we also observe that shNpas4 in D2-Cre rats has no effect on extinction day 1 behavior or cocaine primed reinstatement. These findings suggest that “motivation” for drug-seeking is not altered, but that the drug-cue association is disrupted. However, it would be interesting in the future to examine progressive ratio to ascertain possible influences of NPAS4 in D2-cells on the motivation to work for cocaine.

The authors found that knockdown of Npas4 in D2R-MSNs significantly reduced the number of presumed FOS-positive neurons in D1-MSNs (Fig. 4D, right). The authors need to explain why knockdown of Npas4 in D2R-MSNs reduces FOS-positive neurons in D1R-MSNs. Does the increase of neural activity in D2R-MSN caused by ShNpas4 affect neural activity in the D1R-MSN? The authors need to prove this experimentally.

The reviewer raises an interesting point that we did not test in the current study. The ratio of FOS+ D1:D2 cells is altered by D2-shNPAS4, but the ratio change doesn't mean that there was a reduction in FOS+ D1 cells. We have now evaluated the FOS+ cell number explicitly. There is a significant interaction between shNPAS4 and cell type, but none of the post hoc tests are significant. That said, the increase in FOS+ D2-MSNs is a statistical trend ($p=0.053$), which appears to be a major driver of the D1:D2 FOS+ ratio. There is also a decrease in putative D1-MSN FOS+ number, but it is not statistically different from control ($p=0.4$). We now include this new analysis in the revised manuscript, and we discuss that the ratio shift is likely due to the recruitment of more activated D2-MSNs in the absence of NPAS4 in this population rather than a reduction in D1-MSN activation.

In Fig. 4, when Npas4 is knocked down using D1R-Cre mice, it is incorrect to describe the cells that were not knocked down as D2R-MSN. Similarly, when Npas4 is knocked down using D2R-Cre mice, it is incorrect to describe the cells that were not knocked down as D1R-MSN. The authors should state this correctly, as other subtypes of cells may be included.

The reviewer is correct. MSNs are abundant compared to other cell types in the striatum, so we expect them to be the main population driving the result. We have changed our language in the text and figure to say “putative” D1- or D2-MSN, as these

examined cells are either non-D1 or non-D2 expressing.

The authors measured the number of Fos-positive cells in dorsolateral VP in Fig. 4. To show direct evidence the accumbal D2-MSNs to dorsolateral VP pathway was regulated by Npas4 in D2-MSN. Connectivity between accumbal D2-MSNs and Fos-positive neuron in dorsolateral VP of shScram_D2 and ShNpas4_D2 should be monitored by tracing technique.

The dorsolateral ventral pallidum is a well-established projection target of NAc D2-MSNs. Regardless of whether the change in dVP Fos is due to direct or indirect effects of shNpas4 on D2-MSN activity levels, we feel this experiment demonstrates the important point that loss of Npas4 in NAc D2-MSNs impacts a known downstream brain region of the NAc that is activated and required for cocaine seeking-related behavior. We have revised the language of our findings to be more accurate and temper our conclusions of the experiment.

The authors found that Npas4 knockdown in mice conditioned to cocaine, but not saline, produced a significant increase in D2-MSN dendritic spine density in Fig. 6B and C. Lissek et al. have recently reported that Npas4 knockdown decreased spine density under basal conditions and prevented the cocaine-evoked increase in dendritic spine density in D1R-MSN. Whereas there were no significant differences in spine density in any treatment group in D2R-MSNs, indicating that Npas4-mediated regulation of spine density is restricted to D1-MSNs (EMBO Reports. 2021 Dec 6;22(12):e51882. doi: 10.15252/embr.202051882.). The authors should resolve this critical discrepancy in the main text of revised manuscript.

A key difference between our study and Lissek et al is that they utilized a pan-neuronal shNpas4, whereas we employed a cell type-specific shRNA approach. It's possible that pan-neuronal loss of Npas4 produces a distinct change in spine density than when Npas4 is manipulated in only D2+ cells. It's possible that pan-neuronal Npas4 knockdown produces distinct effects on D2-MSN spine density produced by compensatory or indirect effects. To address this difference, we have modified the discussion section to point out these differences and potential reasons for the differences.

Although almost all experiments in this study have been carried out with a single knockdown vector, the use of multiple siRNAs against the same target, experiments to rescue RNAi effects by expressing siRNA-resistant genes or the use of knockout mice are required to increase the reliability of the data.

We have designed and validated a second, sequence-independent shRNA and have replicated our finding that knockdown of *Npas4* in NAc D2+ neurons significantly decreases cocaine CPP behavior. Moreover, in our original report (Taniguchi et al, 2017), we used both floxed NPAS4 mice + NAc injected AAV2-Cre to show the same effect on cocaine CPP as the single Npas4 shRNA (same used in this study). We feel

that the addition of this new data plus the prior replication using floxed Npas4 mice, provides further confidence in the role of NPAS4 in NAc D2-MSNs.

Minor point

It is unclear the injected position of each animal. In explaining AAV experiments the authors should refer more accurately to the site of injection. The authors should also provide a composite drawing of the injection sites based on fluorescence in their slices in the supplement in order to validate a consistent injection site.

We have added more detail to the methods and now include a supplemental figure displaying our injection sites based on viral fluorescence as the reviewer requested.

In Fig 3A, which animal did the authors use, mice or rats? The authors describe mice in the main text, but rats in Figure legend.

We use mice for the CPP experiments and rats for the IVSA experiments. We have clarified this in both the text and the figure legend.

The authors described Penk and Tac1 in the main text. There was no figure panel regarding Penk and Tac1 in Fig. 5e.

We have added the figure panels for Penk and Tac1.

The authors need to describe the measurement time of the CPP post-test in Materials and Methods section.

We have clarified the time for the CPP post-test in the Methods section.

For the graph of CPP scores, not only the post-test values should be shown, but also the pre-test values.

We now include all CPP pre-test scores in the respective supplemental figure for each experiment.

As with other Figs, the graphs in Figs 4c, 4d, 4h, 4i and 6c should be presented as bar and scatter plots.

We have made the suggested changes and now include the scatter plots throughout.

Reviewer #3 (Remarks to the Author):

The study described in this manuscript is a tour de force. It identifies a host of new properties of the transcription factor NPAS4 and how these properties play a role in cocaine-induced changes in brain and behavior. The study makes use of a wide variety of powerful approaches to address key issues in NPAS4 activation (e.g. when, where and the downstream consequences for neuronal activity and behavior), including the

introduction and validation of an NPAS4-TRAp2a mouse line that should be very useful for future studies in the field. The results in the manuscript bring up lots of new and interesting questions that presage some exciting work ahead. I have several concerns about the manuscript in its present form, each of which can be readily addressed.

1. Although it is cited, there is scant discussion (unless I somehow missed it?) of the Lissek et al. 2021 study examining NPAS4's role in shaping neuronal excitability in accumbens MSNs and cocaine-induced behavior as it relates to the current study. This is warranted, especially as the results appear to be slightly at odds with those in the current study?

We thank the reviewer for catching this oversight and now include additional discussion of Lissek et al., 2021, particularly with consideration of noted discrepancies mentioned above.

2. The study uses both female and male mice. This is as it should be, of course. At the same time there seems to be no statement of whether the data were examined with respect to sex and whether any differences were noted. The study may not have been powered to detect subtle differences but given that the data in this manuscript will likely set the stage for a wide array of future studies on the topic and that the literature is now replete with data indicating sex-differences in neural factors that contribute to reward processing and behavior, this should be addressed in some way.

Our analysis does not suggest the presence of sex differences in our studies, but the reviewer is likely correct in that we may be underpowered to detect subtle differences. We now include statements where appropriate about how the data were analyzed with respect to sex and have disaggregated our data by sex where feasible in order to increase transparency.

3. (Minor) The title should include "cocaine" as this was the drug used to test the cue- and relapse-like-related behaviors and different drugs can engage different neural elements.

Good point! We have adjusted the title accordingly.

4. (Minor) There are quite a few typos throughout the manuscript (e.g. repeated words, incorrect tenses, etc.) that can get cleaned up.

We thank the reviewer for pointing this out and have edited the manuscript accordingly.

REVIEWER COMMENTS

Reviewer #1 (Remarks to the Author):

In this revision, the authors have done significant additional work to address the concerns of the reviewers. In particular, the addition of the second independent Npas4 shRNA, the more detailed analyses of Npas4 expression using multiple methods, and the addition of discussion regarding other papers in the literature with different experimental outcomes make this a rigorous presentation.

I did find a few points that still need to be fixed, however.

1.

[REDACTED]

2. I still find the self-administration data to be less compelling than the other data, though they are presented here mostly as complementary to the mouse studies. Re-reading, there are two points I missed before, so I am adding them here not to raise new concerns, but to offer a way the authors can solidify the value of these data in their study.

First, I did not see evidence of validation of the shRNA-mediated knockdown in the rat. I assume the sequences are not the same for mouse and rat shRNAs, though also I did not see where the shRNA sequences were in the methods (these should be provided somewhere).

Second, looking at the data in Figure 3H (D1) and 3I (D2), it seems like the variability in the data in the D1 rats would likely preclude the ability to see an effect of the size observed in the D2 rats. This does not undermine the observation that there is an effect in the D2 rats, but I suspect the experiment is underpowered for the authors to state that they see no effects in the D1 rats. A power analysis would reveal how many rats would be needed to conclude there is no effect in D1. I think presentation of these results as they stand as an extension of the D2 data in mouse could be fine as long as the authors state carefully what they can and cannot say with statistical backing about the D1 rats.

3. The new shRNA in Figure S30 nicely recapitulates the CPP data from the other shRNA in Figure 3D. What is most interesting is that CPP is not just lost with the knockdown – it actually looks to be aversive in many of the mice. This could just be variability but it is quite striking when looking at the independent experiments. If the authors think this is an interesting effect they may want to comment on it.

4. Thank you for clarifying the FANS method. The manuscript never explains what the abbreviation 7AAD stands for or why it is used, so this should be added.

5. The authors have included much more data about the cell types that express Npas4 and they are being very transparent here about how the different methods (snRNAseq, in situ, immunostaining, transgenes) give somewhat different quantitative results for both Npas4 expression and expression of the marker genes. I agree the authors needed to demonstrate that they have significant Npas4 in MSNs given that this contrasts with their prior paper. However the text is a bit over stating the MSN-centric expression. Here are some examples the authors might want to adjust:

Line 132: “We observed that 96% of NAc NPAS4-TRAPed neurons expressed either Drd1a (49%) or Drd2 (27%), indicating that these cells were predominantly MSNs”. These data were obtained by in situ for Drd1a and Drd2. NAc interneurons are well established to express DA receptors as well, and even though these are at lower levels than MSNs, they are likely detected by in situ. Unless the authors use additional marker genes, they cannot say that Drd1a/Drd2+ cells are necessarily MSNs.

Line 137: "Using D1-td-Tomato x D2-eGFP reporter mice and immunohistochemistry, we observed that cocaine conditioning induced NPAS4 protein-positive neurons predominantly in NAc D1- and D1-MSNs". Only 49% of the total NPAS4+ cells are overlapping the reporters, so this is not "predominantly". Also, unless someone has done double labeling to confirm that the transgene reporters in these lines do not overlap interneuron markers, the concern above still applies.

Line 387: "...~75% of cocaine-conditioned TRAP cells in the NAc are D1- or D2-MSNs (~50% D1 and ~25% D2)." They are D1 and D2+ yes, but not necessarily MSNs.

6. The snRNA sequencing data in Figure 2 and Figure 5 are strikingly different in the percentages of different types of cells they show – specifically the >20% interneurons in Figure 2 is quite unlike any estimation of this population fraction I have seen before and very unlike Figure 5. These data highlight the challenging variability of cell estimations from single cell data. There is also a lot of discussion in the literature about the most robust statistical methods to quantify DEGs in single cell data. Because of the way the authors use the single nucleus data in this story, I do not think the quantitative challenges are a major concern, except when the text tries to push heavily on quantitative statements about these data. For example, in line 158 about the distribution of Npas4+ cells matching this overall distribution of cell types, when this appears oddly interneuron biased, and the idea of using quantitative language like 1% versus 2% of NPAS4+ cells in line 166 being meaningful and the words "vast majority" regarding NPAS4 in MSNs in line 166.

7. About the characterization of FOS+ cells. Line 227: "We injected Cre-dependent shScram or shNpas4 into the NAc of D1- and D2-Cre mice and observed that ~60% of FOS-positive NAc cells in control shScram animals were either D1-MSNs (Figure 4C, left) or putative D1-MSNs (Figure 4D, left) while ~40% of FOS-positive neurons were either D2-MSNs (Figure 4D, left) or putative D2-MSNs (Figure 4C, left), as expected from prior literature". As one of the other reviewers commented, these Cre lines are not perfect, so cells not overlapping D1-Cre are not necessarily D2, which is why the authors added "putative". However In addition to MSNs, FOS is robustly induced by cocaine in PV+ GABAergic interneurons. There are a small number of these cells in the NAc, but given that a significant percentage (up to 30% in some reports) can show FOS induction after drugs like cocaine. Thus for these two reasons, it is not reasonable to call all non-D1 transgene cells "putative D2-MSNs" and vice versa in Figure 4 and the associated text. The more conservative language would be simply to say that cells are D1-Cre+ versus D1-Cre-, with the reverse for the D2-Cre.

8. The Funahashi paper is mentioned in the text but not yet added to the reference list. Also although the Funahashi and Lissek papers are now added to the discussion, the authors don't offer too much in the way of suggestions why the results matter. This would help the readers and I am sure, despite line 425, that the authors do have some ideas of why this may differ.

9. The font is exceptionally small and seems to be grainy in many of the supplementary figures.

Reviewer #2 (Remarks to the Author):

The authors have responded appropriately to my previous comments.

Reviewer #3 (Remarks to the Author):

This is a comprehensive and rigorous study on an interesting and important topic. The extensive revisions have greatly improved the manuscript. I have no remaining concerns.

Response to Reviewers

Hughes, Huebschman et al.

04/19/2024

We thank the reviewers for their continued consideration of our manuscript on NPAS4's role in supporting cocaine-cue associations and relapse-like behavior. We are submitting a revised version which we feel reasonably addresses the remaining points from Reviewer #1. Copied below are reviewer comments and our responses in blue. Corresponding changes made in the manuscript itself are depicted with highlighted text.

Reviewer #1 (Remarks to the Author):

In this revision, the authors have done significant additional work to address the concerns of the reviewers. In particular, the addition of the second independent Npas4 shRNA, the more detailed analyses of Npas4 expression using multiple methods, and the addition of discussion regarding other papers in the literature with different experimental outcomes make this a rigorous presentation.

I did find a few points that still need to be fixed, however.

1.

[REDACTED]

2. I still find the self-administration data to be less compelling than the other data, though they are presented here mostly as complementary to the mouse studies. Re-reading, there are two points I missed before, so I am adding them here not to raise new concerns, but to offer a way the authors can solidify the value of these data in their study.

First, I did not see evidence of validation of the shRNA-mediated knockdown in the rat. I assume the sequences are not the same for mouse and rat shRNAs, though also I did not see where the shRNA sequences were in the methods (these should be provided somewhere).

Second, looking at the data in Figure 3H (D1) and 3I (D2), it seems like the variability in the data in the D1 rats would likely preclude the ability to see an effect of the size observed in the D2 rats. This does not undermine the observation that there is an effect in the D2 rats, but I suspect the experiment is underpowered for the authors to state that they see no effects in the D1 rats. A power analysis would reveal how many rats would

be needed to conclude there is no effect in D1. I think presentation of these results as they stand as an extension of the D2 data in mouse could be fine as long as the authors state carefully what they can and cannot say with statistical backing about the D1 rats.

We thank the reviewer for these suggestions.

First, the sequence for the original shNpas4 targets a region that is identical between mouse and rat and it works in both species. We have added the sequences for both the original shRNA and the second shRNA used to validate our CPP findings to the methods section.

Second, in the D2-Cre rats, we observe a 0.85 effect size (f) on the significant interaction (session x virus) depicted in Figure 3I. When we conduct a power analysis with this effect size, we determine that a total sample size of 6 animals is required to achieve a power of $\beta = 0.85$. In the D1-Cre rats (Figure 3H), we observe an effect size of 0.132 on the non-significant interaction and determine, with this effect size, that a total sample size of 132 animals would be required to achieve a power of $\beta = 0.85$. We have added commentary in the text regarding the statistical backing of our findings.

3. The new shRNA in Figure S30 nicely recapitulates the CPP data from the other shRNA in Figure 3D. What is most interesting is that CPP is not just lost with the knockdown – it actually looks to be aversive in many of the mice. This could just be variability but it is quite striking when looking at the independent experiments. If the authors think this is an interesting effect they may want to comment on it.

Yes, we do think this is an interesting effect! We have added some commentary on this point to the discussion section.

4. Thank you for clarifying the FANS method. The manuscript never explains what the abbreviation 7AAD stands for or why it is used, so this should be added.

We thank the review for catching this oversight and have added methodological details related to the use of 7AAD.

5. The authors have included much more data about the cell types that express Npas4 and they are being very transparent here about how the different methods (snRNAseq, in situ, immunostaining, transgenes) give somewhat different quantitative results for both Npas4 expression and expression of the marker genes. I agree the authors needed to demonstrate that they have significant Npas4 in MSNs given that this contrasts with their prior paper. However the text is a bit over stating the MSN-centric expression. Here are some examples the authors might want to adjust:
Line 132: “We observed that 96% of NAc NPAS4-TRAPed neurons expressed either Drd1a (49%) or Drd2 (27%), indicating that these cells were predominantly MSNs”. These data were obtained by in situ for Drd1a and Drd2. NAc interneurons are well established to express DA receptors as well, and even though these are at lower levels than MSNs, they are likely detected by in situ. Unless the authors use additional marker genes, they cannot say that Drd1a/Drd2+ cells are necessarily MSNs.

Line 137: “Using D1-td-Tomato x D2-eGFP reporter mice and immunohistochemistry, we observed that cocaine conditioning induced NPAS4 protein-positive neurons predominantly in NAc D1- and D1-MSNs”. Only 49% of the total NPAS4+ cells are overlapping the reporters, so this is not “predominantly”. Also, unless someone has done double labeling to confirm that the transgene reporters in these lines do not overlap interneuron markers, the concern above still applies.

Line 387: “...~75% of cocaine-conditioned TRAP cells in the NAc are D1- or D2-MSNs (~50% D1 and ~25% D2).” They are D1 and D2+ yes, but not necessarily MSNs.

We thank the reviewer for pointing this out and agree. We have made the suggested changes.

6. The snRNA sequencing data in Figure 2 and Figure 5 are strikingly different in the percentages of different types of cells they show – specifically the >20% interneurons in Figure 2 is quite unlike any estimation of this population fraction I have seen before and very unlike Figure 5. These data highlight the challenging variability of cell estimations from single cell data. There is also a lot of discussion in the literature about the most robust statistical methods to quantify DEGs in single cell data. Because of the way the authors use the single nucleus data in this story, I do not think the quantitative challenges are a major concern, except when the text tries to push heavily on quantitative statements about these data. For example, in line 158 about the distribution of Npas4+ cells matching this overall distribution of cell types, when this appears oddly interneuron biased, and the idea of using quantitative language like 1% versus 2% of NPAS4+ cells in line 166 being meaningful and the words “vast majority” regarding NPAS4 in MSNs in line 166.

Yes, the reviewer makes a good point regarding the interneuron population in this experiment. The high percentage of “interneurons” seems to largely be driven by a relatively high number of Pnoc+ neurons, which are present (in varying amounts) in both of our snRNA-seq experiments. Pnoc is highly expressed in the BNST and it’s possible that this cluster arises from BNST contamination during the dissection process, explaining the variability between experiments (and experimenters!). We have added commentary addressing this point.

We feel that, while these quantitative statements certainly should not stand alone, they do aid in pulling some meaning from our sequencing data in the context of our other experiments. We understand the reviewers concern, however, and have tempered our language in some areas to address this.

7. About the characterization of FOS+ cells. Line 227: “We injected Cre-dependent shScram or shNpas4 into the NAc of D1- and D2-Cre mice and observed that ~60% of FOS-positive NAc cells in control shScram animals were either D1-MSNs (Figure 4C, left) or putative D1-MSNs (Figure 4D, left) while ~40% of FOS-positive neurons were either D2-MSNs (Figure 4D, left) or putative D2-MSNs (Figure 4C, left), as expected from prior literature”. As one of the other reviewers commented, these Cre lines are not perfect, so cells not overlapping D1-Cre are not necessarily D2, which is why the

authors added “putative”. However In addition to MSNs, FOS is robustly induced by cocaine in PV+ GABAergic interneurons. There are a small number of these cells in the NAc, but given that a significant percentage (up to 30% in some reports) can show FOS induction after drugs like cocaine. Thus for these two reasons, it is not reasonable to call all non-D1 transgene cells “putative D2-MSNs” and vice versa in Figure 4 and the associated text. The more conservative language would be simply to say that cells are D1-Cre+ versus D1-Cre-, with the reverse for the D2-Cre.

The reviewer makes a valid point. We have adjusted the language in the text and figure as suggested.

8. The Funahashi paper is mentioned in the text but not yet added to the reference list. Also although the Funahashi and Lissek papers are now added to the discussion, the authors don't offer too much in the way of suggestions why the results matter. This would help the readers and I am sure, despite line 425, that the authors do have some ideas of why this may differ.

We thank the reviewer for catching this oversight and have added Funahashi et al to the reference list. We have also added additional commentary on this in the discussion section. See highlighted text for changes/additions. We do not have a good explanation for the different results in our study vs. Funihashi *et al* other than the noted differences in study design, and our speculation that overexpression of NPAS4 DNA binding domain alone might produce a gain-of-function effect in D1-cells.

9. The font is exceptionally small and seems to be grainy in many of the supplementary figures.

We have adjusted the font and now include higher resolution images in the supplementary figure file.

Reviewer #2 (Remarks to the Author):

The authors have responded appropriately to my previous comments.

Reviewer #3 (Remarks to the Author):

This is a comprehensive and rigorous study on an interesting and important topic. The extensive revisions have greatly improved the manuscript. I have no remaining concerns.